# Correlating and Predicting Human Evaluations of Language Models from Natural Language Processing Benchmarks

## Abstract

The field of natural language processing (NLP) historically evaluated language models using benchmarks with automated metrics. However, the recent advent of highly capable chat language models (LMs) has caused a tectonic shift from NLP benchmarks to human evaluations. The relationship between these two evaluation processes is unclear and underexplored for chat LMs. Broadly, to what extent are human evaluations and NLP benchmarks correlated with one another? How well can computationally inexpensive and automated benchmarks predict expensive and time-intensive human evaluations? Which benchmarks provide predictive signals for human preference for LMs? What role, if any, should benchmarks play in the era of chat LMs? To answer these questions, we conducted a large-scale study of the relationships between human evaluations and benchmarks. We show that benchmarks are broadly highly correlated with human evaluations, and we identify which benchmarks exhibit strong correlations with human evaluations and which do not. Having established that reliable correlations exist, we fit models to predict a language model's human evaluation scores from its academic evaluation scores and provide evidence that such predictive models can generalize across LM scales.

## 1 Introduction

For decades, the field of natural language processing (NLP) has relied on academic benchmarks and automated metrics (e.g., Accuracy, Brier Score (Brier, 1950), BLEU Papineni et al. (2002)) to evaluate the performance of language models (LMs). These NLP benchmarks provide a standardized and efficient way to measure model capabilities such as machine translation, text summarization, and question answering (Wang et al., 2018; 2019; Srivastava et al., 2022; Gao et al., 2023; Wang et al., 2023a). However, the recent emergence of highly capable chat LMs such as GPT (Ouyang et al., 2022; Achiam et al., 2023), Llama (Touvron et al., 2023a;b; Dubey et al., 2024), Gemini (Team et al., 2023; Reid et al., 2024) and Claude (Anthropic, 2023) has prompted a re-evaluation of how we assess LMs, with a growing emphasis on assessing LMs based on their ability to interact with and assist human users in real-world scenarios (Zheng et al., 2023; Reuel et al., 2024).

This shift towards human evaluations raises important questions about the relationship between NLP benchmarks and human evaluations of chat LMs. Additionally, human evaluations are not without challenges; they can be expensive, time-intensive and noisy, in contrast with computationally cheaper, faster and precise benchmarks. In this paper, we aim to explore the relationship between human evaluations and NLP benchmarks in pursuit of understanding what role, if any, benchmarks should play in the era of chat LMs. As shown in Fig. 1, we seek to answer two key research questions:

1. To what extent are human evaluations and NLP benchmarks correlated with one another?
2. How well can NLP benchmarks predict human evaluations?

To answer these questions, we conducted a large-scale study comparing human evaluations and NLP benchmarks using four Llama 2 Chat language models (LMs) (Touvron et al., 2023b). For human evaluations, we constructed a large-scale dataset of single-turn and multi-turn prompts from a diverse taxonomy (Fig. 2) and collect high quality pairwise preference data of the four Chat Llama 2 models against GPT 3.5 (Ouyang et al., 2022) from paid human annotators. For NLP benchmarks,

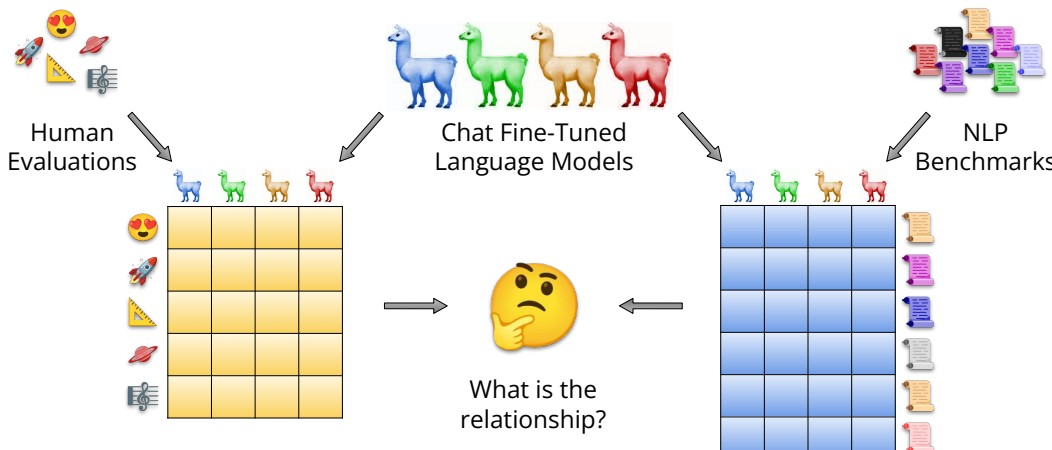

Figure 1: **Correlating and Predicting Human Evaluations of Language Models from Natural Language Processing (NLP) Benchmarks.** We evaluate chat language models on conversational tasks with human pairwise evaluations and on standard NLP benchmarks with automated metrics, then study whether scores on computationally inexpensive and fast NLP benchmarks are correlated with and predictive of expensive and time-intensive human evaluations?

we evaluate the same four Chat Llama 2 models on standard NLP benchmarks under established evaluation processes (metrics, prompting, 0-shot/few-shot, etc.). We analyze pairwise correlations between NLP benchmark and human evaluations to identify which NLP benchmarks correlate highly with human evaluations and which do not. We also aim to identify which human evaluations, if any, are uncorrelated with any NLP benchmarks. We then pivot to predicting human evaluations from NLP benchmarks using overparameterized linear regressions and leave-one-out cross-validation. We investigate the extent to which NLP benchmarks can predict human evaluations.

## 2 RELATED WORK

The evaluation of language models has a rich and constantly evolving history. Human evaluations have long been considered the gold standard (Gatt & Krahmer, 2018; Van Der Lee et al., 2019; Celikyilmaz et al., 2020; Roller et al., 2020; van der Lee et al., 2021), despite serious objections raised regarding the collection, analysis, and interpretation of human evaluation scores (Novikova et al., 2018; Howcroft et al., 2020; Bowman & Dahl, 2021; Karpinska et al., 2021; Clark et al., 2021; Smith et al., 2022; Gehrmann et al., 2023; Finch et al., 2023). Many classic NLP benchmark metrics, such as BLEU (Papineni et al., 2002), NIST (Doddington, 2002), ROUGE (Lin, 2004), and METEOR (Banerjee & Lavie, 2005), were introduced on the premise that they correlate with human judgments. However, subsequent studies revealed that the relationship between automated metrics and human evaluations is often complex and not straightforward (Liu et al., 2016; Novikova et al., 2017; Reiter, 2018; Karpinska et al., 2021). Another prominent class of evaluation methods are based on machine learning models, e.g., word mover distance (Kusner et al., 2015) and BERT-Score (Zhang et al., 2019) that have since evolved into using chat LMs themselves as evaluators (Wang et al., 2023b; Zheng et al., 2024; Chiang & yi Lee, 2023; Chan et al., 2023; Bavaresco et al., 2024; Fu et al., 2024), albeit with limitations, e.g., (Dorner et al., 2024; Szymanski et al., 2024; Thakur et al., 2024).

The earliest investigations into the general relationship between NLP benchmark scores and human evaluations date back to Bangalore et al. (2000), Belz & Reiter (2006), and Liu et al. (2016). In the context of natural language generation, Clinciu et al. (2021) found that embedding-based automated metrics (e.g., BERT-Score (Zhang et al., 2019) and BLEURT Sellam et al. (2020)) correlate more strongly with human judgments compared to word-overlap metrics (e.g., ROUGE (Lin, 2004) and BLEU (Papineni et al., 2002)). In the domain of natural language inference, Schuff et al. (2021) found that automated metrics do not appear to correlate with human judgment scores. However, the majority of these works predate the current era of chat LMs, which exhibit significantly more advanced capabilities compared to their predecessors. This new era motivates our work to investigate the relationship between NLP benchmarks and human evaluations when evaluating chat LMs.

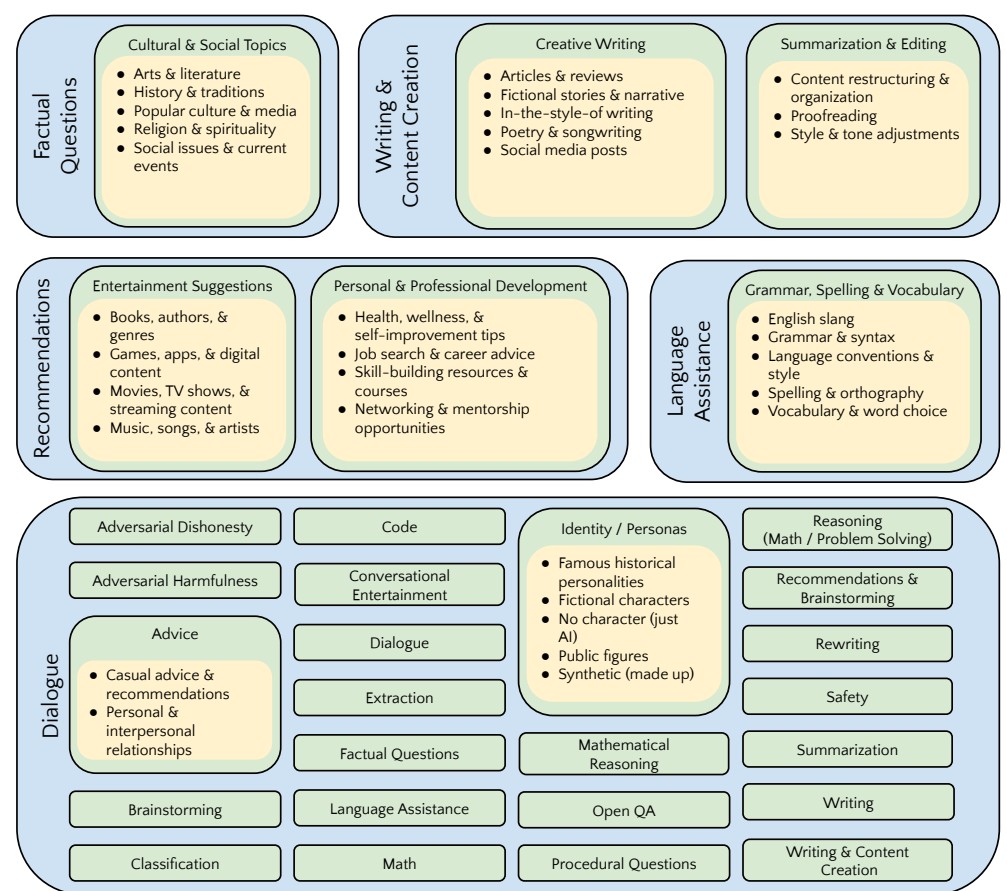

Figure 2: **Human Evaluations: Taxonomy of Single-Turn and Multi-Turn Conversations.** Single-turn and multi-turn prompts were created in a hierarchical taxonomy of 9 areas (blue), categories (green) and subcategories (yellow). Chat Llama 2 generations were then rated against ChatGPT generations by paid human annotators on a 7 point Likert scale (Likert, 1932).

## 3    METHODS: MODELS, HUMAN EVALUATIONS AND NLP BENCHMARKS

We briefly outline our methodology here; for additional information, please see Appendix A.

**Models**    Our paper leverages the Llama 2 model family, consisting of four Chat LMs with 7, 13, 34, and 70 billion parameters pre-trained on 2 trillion tokens and finetuned using supervised finetuning (Sanh et al., 2021; Chung et al., 2022; Longpre et al., 2023) and reinforcement learning from human feedback (Christiano et al., 2017; Ziegler et al., 2019; Stiennon et al., 2020). We chose the Llama 2 models because at the time we collected our data, the Llama 2 family contained leading open-access chat-finetuned models spanning multiple scales with minimal variations in architecture, ensuring consistency in our analyses and a robust foundation for our investigations.

**Human Evaluations: Single Turn & Multi-Turn**    In this work, our aim was specifically to identify which NLP benchmark scores are predictive of human preferences on open-ended prompts representative of real-world chat model usage. We chose this approach to maximize the ecological validity and generalizability of the findings to real-world use cases. For a concrete example, we may want our chat language models (LMs) to excel at providing bespoke career advice; which NLP benchmarks provide useful signals for whether models are improving at such tasks?

To answer such questions, we created a taxonomy of single-turn and multi-turn interactions (Fig. 2) between chat LMs and humans. For single-turn interactions, we generated a diverse set of prompts

spanning common areas of interest: Factual Questions, Procedural Questions, Language Assistance, Writing & Content Creation, Dialogue, Code, Reasoning, Recommendations / Brainstorming and Safety, with nested categories and subcategories. For multi-turn prompts, non-annotator humans were asked to have conversations (3 to 15 turns long) with all models on similar topics of interest: Factual Questions, Procedural Questions, Language Assistance, Writing & Content Creation, Summarization & Editing, General Dialogue, Reasoning and Recommendations / Brainstorming. This taxonomy was chosen to broadly cover common use-cases of Chat LMs. Example prompts include: "What is the tallest mountain in the world?" (Factual Question); "How do I make minestrone soup?" (Procedural Question); "Please make this sentence more friendly: I need you to stop parking in my space" (Language Assistance); "Write me a poem about getting to the weekend after a long day at work" (Writing & Content Creation). See Appendix A.2 for more information.

We then paid human annotators to evaluate each of the four Chat Llama 2 models against ChatGPT 3.5 (Ouyang et al., 2022) (gpt-3.5-0301) on a dataset of single-turn and multi-turn prompts (Fig 2). We chose gpt-3.5-0301 because, at the time this data was collected, gpt-3.5-0301 was a good balance of three desirable properties for our study: performant, cheap, and stable. For each pair of conversations (one conversation with Chat Llama responses and the other with ChatGPT responses), at least three unique human annotators independently indicated which conversation was preferred using a Likert scale (Likert, 1932) from 1 to 7, where 1 denotes the Chat Llama model was strongly preferred and 7 denotes gpt-3.5-0301 was strongly preferred. Across the 11291 single-turn samples and 2081 multi-turn samples, we had at least 3 unique human annotators per pairwise comparison, with 2104 unique human annotators overall. For our analyses, we averaged the annotators' scores for each pairwise comparison to give us an average human evaluation score per datum.

**Natural Language Processing (NLP) Benchmarks**   We evaluated the four Chat Llama 2 models on large-scale and commonly-used NLP benchmarks: AGI Eval (Zhong et al., 2023), AI2 Reasoning Challenge (ARC; both Easy and Hard) (Clark et al., 2018), BIG Bench Hard (Srivastava et al., 2022; Suzgun et al., 2022) BoolQ (Clark et al., 2019), CommonSenseQA (Talmor et al., 2019), COPA (Roemmele et al., 2011), DROP (Dua et al., 2019), GSM8K (Cobbe et al., 2021), HellaSwag (Zellers et al., 2019), HumanEval (Chen et al., 2021), InverseScaling (McKenzie et al., 2022a;b; 2023), MBPP (Austin et al., 2021), MMLU (Hendrycks et al., 2020), Natural Questions (Kwiatkowski et al., 2019), OpenbookQA (Mihaylov et al., 2018), PIQA (Bisk et al., 2020), QuAC (Choi et al., 2018), RACE (Lai et al., 2017), SIQA (Sap et al., 2019), SQUAD (Rajpurkar et al., 2016), TLDR (Völske et al., 2017), TriviaQA (Joshi et al., 2017), WinoGrande (Sakaguchi et al., 2021) and XSum (Narayan et al., 2018). Some of these benchmarks (e.g., MMLU) contain subsets (e.g., Jurisprudence) that we do not aggregate over. These tasks cover commonsense reasoning, world knowledge, reading comprehension, coding and more. We used standard evaluation processes for all academic benchmarks including prompt formatting, metrics, 0-shot/few-shot, etc. This structured approach facilitates an exhaustive examination of model performances across varied metrics. For more information, see Appendix A.1.

**Scores for Subsequent Analyses**   For each dataset and evaluation process (either human or NLP), we average each model's scores across all samples, yielding two matrices of scores:

$$X_{\text{NLP}} \in \mathbb{R}^{160 \times 4} \qquad X_{\text{Human}} \in \mathbb{R}^{55 \times 4}$$

Here, 4 is the number of models, 160 is the number of NLP benchmarks per model and 55 is the number of human evaluation area-category-subcategory scores per model. We subsequently study the correlations between $X_{\text{NLP}}$ and $X_{\text{Human}}$, then test how well $X_{\text{NLP}}$ can predict $X_{\text{Human}}$.

## 4   CORRELATING HUMAN EVALUATIONS WITH NLP BENCHMARKS

We began by computing correlations between human evaluations and NLP benchmarks, computing three standard correlations over the 4 average scores per model — Pearson (Galton, 1877), Spearman (Spearman, 1904) and Kendall (Kendall, 1938) — giving us three correlation matrices of shape $160 \times 55$ between every pair of NLP benchmark and human evaluation area-category-subcategory (Fig. 3). Pearson correlation measures the linear relationship between two continuous variables, whereas Spearman and Kendall correlations assess the monotonic relationship between two variables; Spearman correlation is based on the rank order of the data points, whereas Kendall correlation is determined by the number of concordant and discordant pairs. By using different correlation metrics, we aim to robustly characterize the relationships between human and NLP benchmarks.

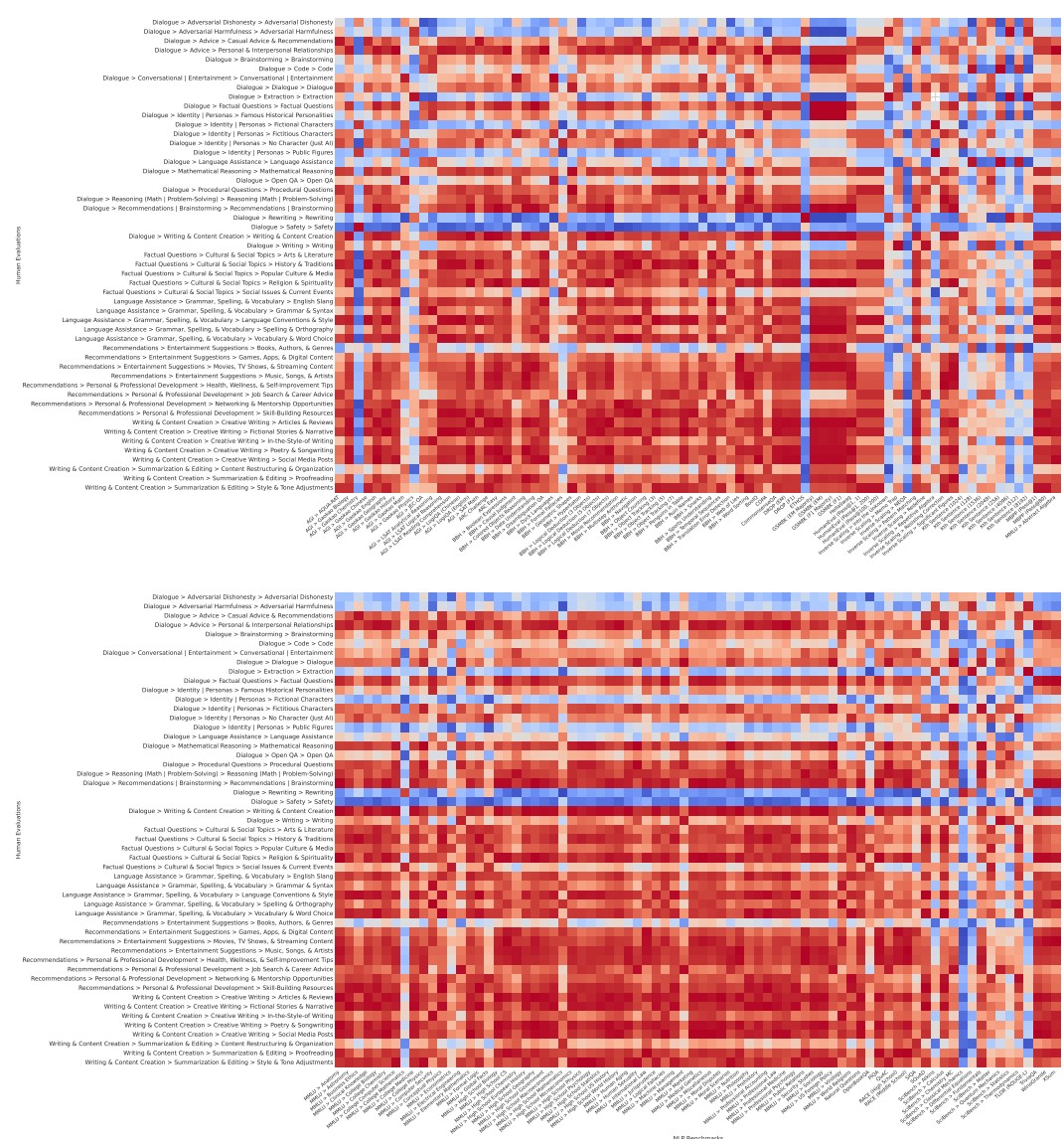

Figure 3: **Pearson Correlations Between Human Evaluations and NLP Benchmarks.** Rows: Human evaluation areas-categories-subcategories. Columns: NLP benchmarks. The heatmap is row-wrapped to fit on the page. Large positive correlations (+1) are shown in red. Large negative anticorrelations (-1) are shown in blue. Low uncorrelations ($\sim$0) are shown in light-white-gray.

Macroscopically, at the most coarse grouping of human evaluations in our taxonomy (i.e., areas) (Fig. 2), we found that average NLP benchmark scores are highly correlated with average human scores for all human evaluation areas under all three correlation metrics (Fig. 4 top). Due to the small number of models ($N = 4$), Spearman and Kendall correlations suffer discretization effects (Fig. 11), inducing an illusion of undulations. These strong correlations suggest that, at a high level, NLP benchmarks are reasonable proxies for human judgments of LM quality.

Mesoscopically, at the level of human evaluation areas and categories, we find that NLP benchmarks remain highly correlated with human evaluations, with two notable types of exceptions (Fig. 4). First, Adversarial Dishonesty, Adversarial Harmfulness, and Safety are anti-correlated with most NLP benchmarks, potentially indicating that these adversarial and safety-focused categories are more easily transgressed by more capable LMs; an alternatively hypothesis could be that safety benchmarks simply are not especially good, as demonstrated by Ren et al. (2024). Second, Language Assistance and Open Question Answering are uncorrelated with most NLP benchmarks, suggesting that these categories may require new NLP benchmarks. Open Question Answering was surprising given that

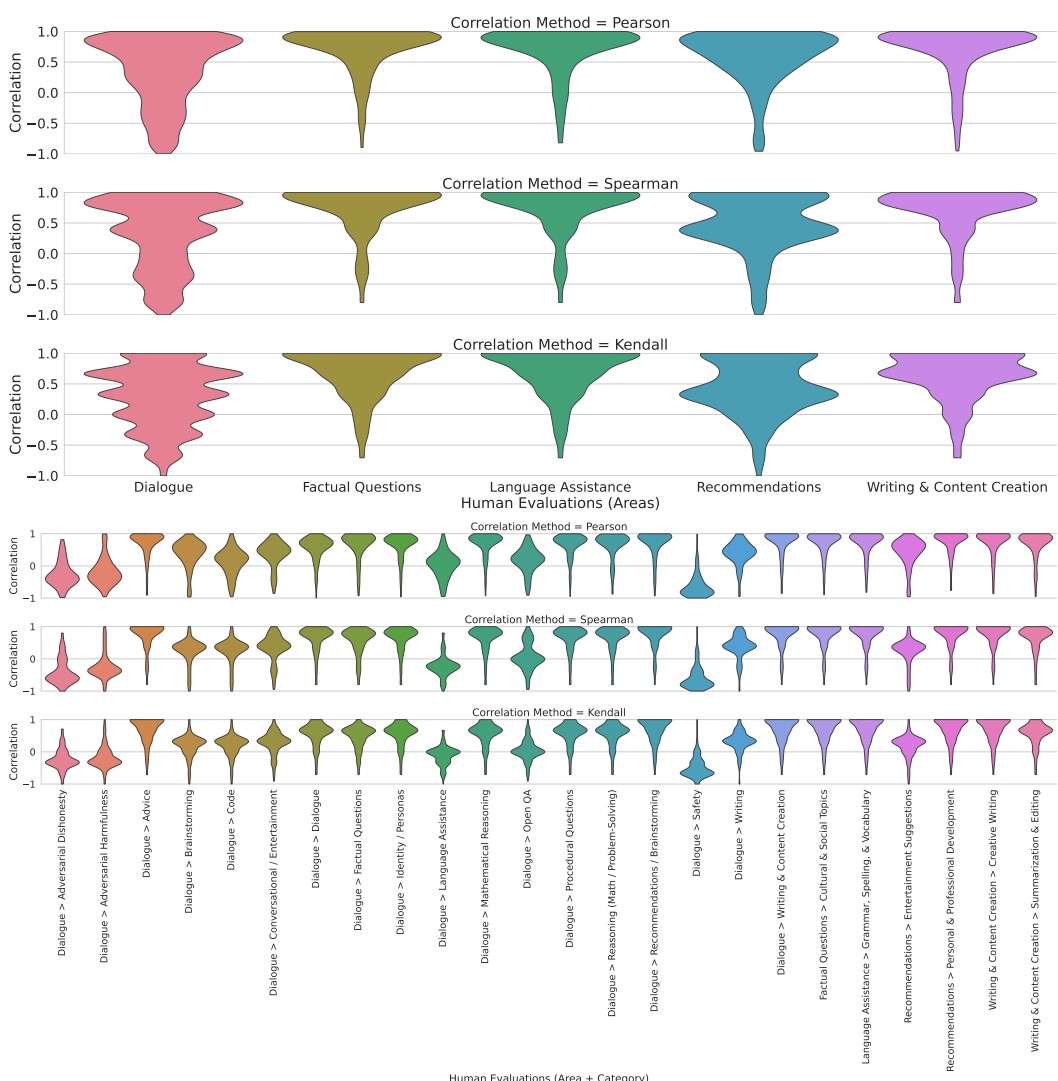

Figure 4: **Distributions of Correlations between Human Evaluations and NLP benchmarks.** Top: Macroscopically, for each human evaluation area, Chat LM scores are typically highly correlated with NLP benchmarks. Bottom: Mesoscopically, human and NLP benchmarks remain positively correlated, with notable exceptions: Adversarial Dishonesty, Adversarial Harmfulness and Safety are anticorrelated with most NLP benchmarks, and Language Assistance and Open QA are uncorrelated.

some of our NLP benchmarks are open question answering datasets, e.g., OpenBookQA (Mihaylov et al., 2018). We found the three correlations metrics visually agreed with one another and were themselves tightly coupled (App. Fig. 11), and so we present only one (Pearson) moving forward, with equivalent plots of the other two (Spearman, Kendall) deferred to the appendix.

### 4.1 WHICH HUMAN EVALUATIONS HAVE FEW-TO-NO CORRELATED NLP BENCHMARKS?

To the best of our ability to discern, none. Every human evaluation seemed to have at least some NLP benchmarks that were either correlated or anticorrelated with it. This result is promising because it suggests human evaluations might be predictable from NLP benchmarks (Sec. 5).

### 4.2 WHICH NLP BENCHMARKS EXHIBIT HIGH CORRELATIONS WITH HUMAN EVALUATIONS?

To answer this question, we ordered NLP benchmarks based on their average correlation score with all of the human evaluation areas, categories and subcategories. We found many NLP benchmarks

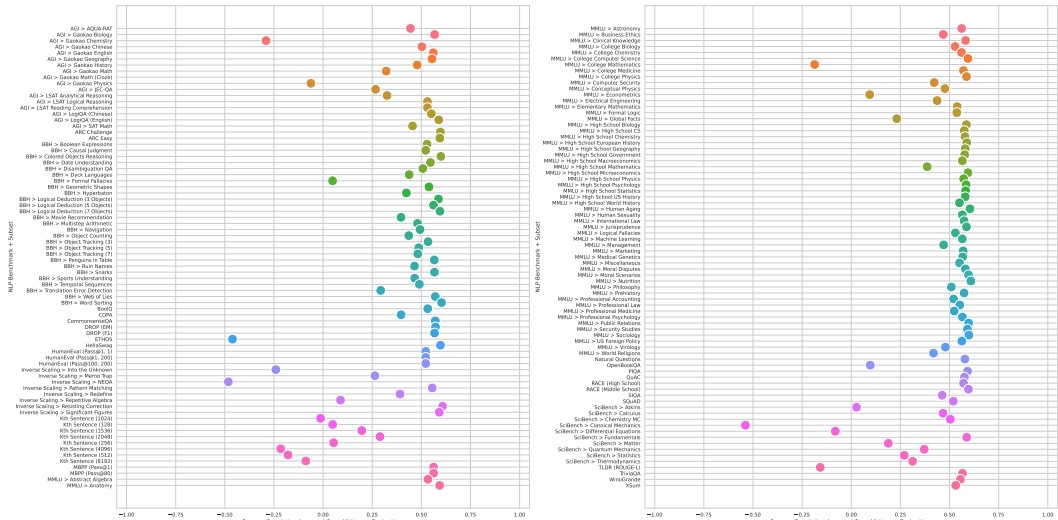

Figure 5: **NLP Benchmarks Ranked by Average Pearson Correlation over All Human Evaluations.** Certain benchmarks have higher correlations with human evaluations, including a subset of MMLU, a subset of BIG Bench Hard, HellaSwag, ARC, RACE, PIQA, NaturalQuestions, QuAC, and CommonSenseQA. Other benchmarks were weakly or uncorrelated with human evaluations: ETHOS, Kth Sentence, Inverse Scaling (with the exception of Resisting Correction Classification), OpenBookQA, COPA, SciBench (with the exception of Fundamentals of Physics) and SIQA.

have high average correlation with human evaluations (Fig. 5); the highest average correlation NLP benchmarks include a subset of MMLU (Nutrition, Human Aging, Sociology, Public Relations, Moral Scenarios, College Computer Science), a subset of BIG Bench Hard (Word Sorting, Reasoning About Colored Objects, Logical Deduction), HellaSwag, ARC, RACE, PIQA, NaturalQuestions, QuAC, CommonSenseQA, DROP and TriviaQA. Other benchmarks were less correlated or uncorrelated with human evaluations: ETHOS, Kth Sentence, Inverse Scaling (with the exception of Resisting Correction Classification), OpenBookQA, COPA, SciBench (with the exception of Fundamentals of Physics) and SIQA. Upon investigating more closely, some of the most highly correlated NLP benchmarks make sense. For instance, Inverse Scaling's Resisting Correction Classification ranked second highest for being correlated with human evaluations, and the task measures a highly desirable capability for human users: the LM's ability to follow user instructions that run counter to the LM's natural inclinations.

### 4.3 WHAT COMMUNITIES EXIST BETWEEN HUMAN EVALUATIONS AND NLP BENCHMARKS?

To detect what communities exist between human evaluations and NLP benchmarks, we computed the singular value decomposition of the pairwise Pearson correlation matrix between human evaluations and NLP benchmarks (Fig. 6 top). The maximum rank the correlation matrix can have is 4 because the correlations are computed over the 4 Chat Llama 2 models, but we found that the correlation matrix has only 3 non-zero singular values (App. Fig. 12). Decomposing the correlation matrix into its 3 rank-one components $\sigma_1 u_1 v_1^T + \sigma_2 u_2 v_2^T + \sigma_3 u_3 v_3^T$ revealed three levels of increasing fine-grained structure in the correlations (App. Fig. 13). We then visualized the human evaluations and NLP benchmarks in the (dimension-scaled) plane defined by the first two rank-one components of the Pearson correlation matrix (Fig. 6 bottom).

The bulk of human evaluations and NLP benchmarks live in one community; however, there are also several smaller interesting communities. Starting on the left of Fig. 6 and moving clockwise, at the top left is a loose community of Dialogue.Code, Dialogue.Language Assistant, several Kth Sentence tasks, Openbook Question Answering (OBQA), AGI.LSAT, AGI.Lawyer Qualification Test, which generally measure model capabilities at identifying and using key information within the context. On the top right, Inverse Scaling.NEQA Classification is alone; this benchmark measures whether models are tripped up by negated questions, which most humans try not to do and likely explains why

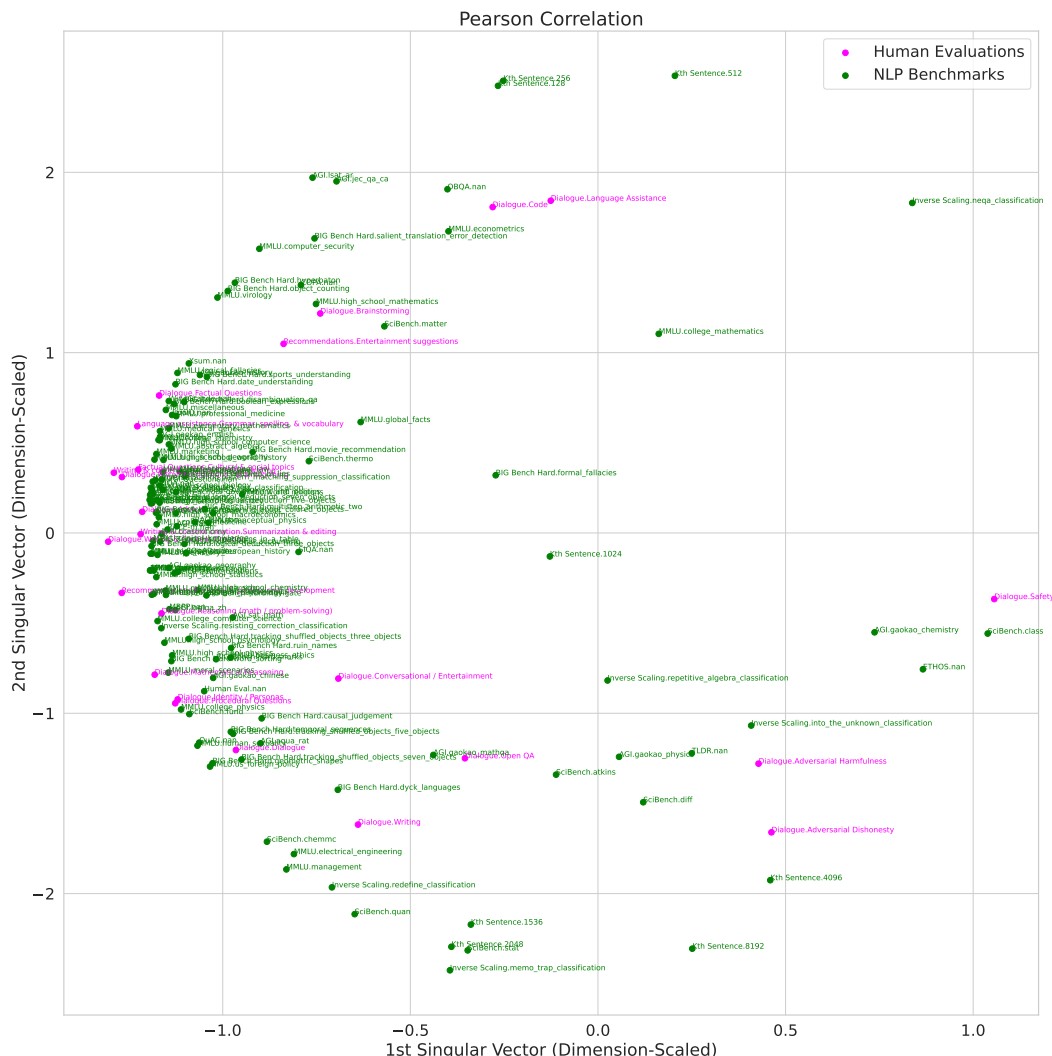

Figure 6: **Structure of Pairwise Pearson Correlations Between Human Evaluations and NLP Benchmarks.** Top: Each row is a human evaluation area and category, and each column is an NLP benchmark and task; values are Pearson correlations ranging from anticorrelated (-1) to correlated (+1). The correlation matrix has 3 non-zero singular values (App. Fig. 12). Bottom: Human evaluations and NLP benchmarks are plotted projected along the (dimension-scaled) first two singular modes of the Pearson correlation matrix. The bulk of evaluations live in one community (left), with smaller communities (top, bottom, right); for an in-depth interpretation, see Sec. 4.3.

this benchmark is isolated. On the right and lower right side, Dialog.Safety is next to ETHOS, a hate speech detection benchmark, and AGI.Gaokao Chemistry, a chemistry benchmark. This community is also close to another community in the lower right comprised of Dialogue.Adversarial Harmfulness, Dialogue.Adversarial Dishonesty, Inverse Scaling.Into the Unknown, TLDR. In the lower left, Dialogue.Open QA and Dialogue.Writing are near BIG Bench Hard's Dyck Languages, Geometric Shapes and Tracking Shuffled Objects and multiple science and factual knowledge benchmarks like MMLU's Electrical Engineering, Management,, SciBench's Quantum Chemistry (quan and chemmc). BIG Bench Hard's Formal Fallacies) and Kth Sentence (1024) lie in the center, disconnected from most other evaluations.

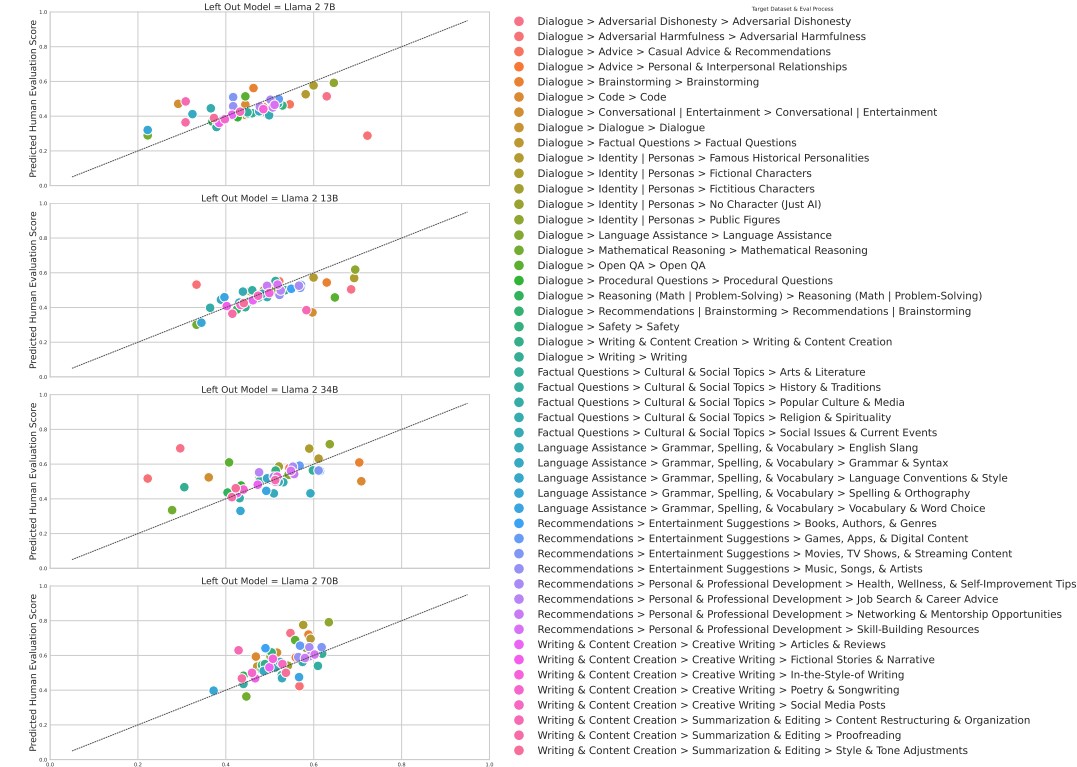

Figure 7: **Leave-One-Out Cross Validated Linear Regression Predictions of Human Evaluations.**
Overparameterized linear regressions typically accurately predict human evaluation scores from all
NLP benchmark scores. Each subfigure shows predicted human evaluation scores against actual
human evaluation scores on each of the four left-out Chat Llama 2 models colored by the particular
area, category and subcategory of human evaluation.

## 5    PREDICTING HUMAN EVALUATIONS FROM NLP BENCHMARKS

Having established the existence of correlations between human evaluations and NLP benchmarks,
we next investigated the feasibility of predicting human evaluations from NLP benchmarks. Our goal
is to build predictive models that accurately predict a language model's average human evaluation
scores per evaluation areas and categories using the model's average scores on NLP benchmarks
and tasks. However, we faced a significant challenge due to the overparameterized nature of our
data: for each target human evaluation area or category, there are approximately 150 covariates (NLP
benchmarks and tasks) but only 4 samples (Chat Llama 2 models).

**Predictive Modeling: Overparameterized Linear Regressions**    To predict human evaluations
from NLP benchmarks, we used overparameterized linear regression. In general, overparameterized
linear regression is known to be capable of generalizing (App. Sec. A.3), although whether linear
models would generalize in this setting was an empirical question. For each human evaluation area
and category, we fit a linear model to predict a language model's average human evaluation score
from its average scores on all NLP benchmarks and tasks. To assess the predictive accuracy of these
overparameterized models, we employed leave-one-out cross validation: we fit four separate linear
models, each time fitting on three of the chat LMS' scores and holding out the fourth to test the
performance of the linear model. This approach allows us to estimate the models' performance
on unseen data, albeit with limitations due to the small sample size. Before fitting the models, we
normalized all human evaluation scores to lie in $[0, 1]$ rather than $[-7, -1]$ (recalling that higher
scores indicate the human evaluator prefers the Chat Llama 2 model compared to GPT-3.5).

**Results**   Across the various human evaluation areas and categories, we found that the linear models' predicted average human evaluation scores generally align well with the actual average human evaluation scores, as evidenced by most points falling close to the identity line in the predicted score vs. actual score plane (Fig. 7). This suggests that, despite the overparameterization, the linear models can capture meaningful relationships between NLP benchmarks and human evaluations. However, we caution against over-interpreting these results, as the small sample size and the assumption of linearity may limit the generalizability of these findings to other language models or evaluation settings.

To gain insight into which NLP benchmarks are most informative for predicting human evaluation scores, we examine the learned weights of the linear models (Fig. 18). NLP benchmarks with consistently high absolute weights across different human evaluation areas and categories are likely to be more predictive of human judgments. However, due to the overparameterized nature of the models, we refrain from drawing strong conclusions about the relative importance of individual benchmarks and instead focus on the overall predictive performance. These results suggest that scaling up the number of chat LMs and human evaluation data could unlock highly predictive models of slow, noisy and expensive but valuable human evaluations using fast, precise and cheaper NLP benchmarks.

## 6 DISCUSSION

In this paper, we explored the relationship between human evaluations and NLP benchmarks of chat-finetuned language models (chat LMs). Our work is motivated by the recent shift towards human evaluations as the primary means of assessing chat LM performance, and the need to understand the role that NLP benchmarks can play in this new era.

Through a large-scale study of the Chat Llama 2 model family on a diverse set of human and NLP evaluations, we demonstrated that NLP benchmarks are generally well-correlated with human judgments of chat LM quality. However, our analysis also reveals some notable exceptions to this overall trend. In particular, we find that adversarial and safety-focused evaluations, as well as language assistance and open question answering tasks, exhibit weaker or negative correlations respectively with NLP benchmarks. We also explored predicting human evaluation scores from NLP evaluation scores using overparameterized linear regression models. Our results suggest that NLP benchmarks can indeed be used to predict aggregate human preferences, although we caution that the limited sample size and the assumptions of our models may limit the generalizability of these findings. Our results suggest that NLP benchmarks can serve as fast and cheap proxies of slower and expensive human evaluations in assessing chat LMs.

Additionally, our work highlights the need for further research into NLP evaluations that can effectively capture important aspects of LM behavior, such as safety, robustness to adversarial inputs, and performance on complex, open-ended tasks. It is possible that new NLP benchmarks can provide signals on these topics, e.g., (Wang et al., 2023a). Of particular interest is developing human-interpretable and scaling-predictable evaluation processes, e.g., (Schaeffer et al., 2024a; Ruan et al., 2024; Schaeffer et al., 2024c). Developing and refining such evaluation methods (Madaan et al., 2024), as well as detecting whether evaluations scores faithfully capture models' true performance (Oren et al., 2023; Schaeffer, 2023; Roberts et al., 2023; Jiang et al., 2024; Zhang et al., 2024; Duan et al., 2024) will be crucial for ensuring that LMs are safe, reliable, and beneficial as they become increasingly integrated into real-world use cases.

In conclusion, our study provides insights into the relationship between human evaluations and NLP benchmarks of chat language models. By leveraging the complementary strengths of both human and NLP benchmarks, we can build a more complete understanding of LM capabilities and behaviors, ultimately enabling the development of models more capable, trustworthy, and beneficial to society.

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

## A  EXPERIMENTAL METHODOLOGY: DATA AND ANALYSES

### A.1  DATA: NATURAL LANGUAGE PROCESSING (NLP) BENCHMARK SCORES

We chose which NLP benchmarks to include based largely on which frontier AI models were reporting performance scores on. Llama 1 (Touvron et al., 2023a) and Llama 2 (Touvron et al., 2023b) were our primary guides, as were Gemini 1 and Gemini 1.5 (Team et al., 2023; Reid et al., 2024), Claude 3 (Anthropic, 2023), and Mistral (Jiang et al., 2023). We evaluated the 4 Llama 2 Chat models following the same evaluation processes reported in the Llama 2 paper (Touvron et al., 2023b). This included matching the prompt formatting, automated metric scoring and (when appropriate) few-shot prompting and chain-of-thought prompting. After evaluating the four Chat Llama 2 models on these NLP benchmarks and evaluation processes, we obtained 160 scores per model for our analyses.

Table 1: **Natural Language Processing Datasets and Evaluation Processes.** In this work, we used the above well-established NLP datasets and evaluation processes. "cot" means Chain-of-Thought prompting (Cobbe et al., 2021; Wei et al., 2022). "Gen" refers generations per evaluation. For more information, please see Section 3 and Appendix A.

| Benchmark | Subset | Metric | Shots | Additional |
|---|---|---|---|---|
| AGI | aqua_rat | Acc Char | 5 | – |
| AGI | gaokao_biology | Acc Char | 5 | – |
| AGI | gaokao_chemistry | Acc Char | 5 | – |
| AGI | gaokao_chinese | Acc Char | 5 | – |
| AGI | gaokao_english | Acc Char | 5 | – |
| AGI | gaokao_geography | Acc Char | 5 | – |
| AGI | gaokao_history | Acc Char | 5 | – |
| AGI | gaokao_mathcloze | Exact Match | 5 | – |
| AGI | gaokao_mathqa | Acc Char | 5 | – |
| AGI | gaokao_physics | Acc Char | 5 | – |
| AGI | jec_qa_ca | Acc Char | 5 | – |
| AGI | jec_qa_kd | Acc Char | 5 | – |
| AGI | logiqa_en | Acc Char | 3 | – |
| AGI | logiqa_zh | Acc Char | 3 | – |
| AGI | lsat_ar | Acc Char | 3 | – |
| AGI | lsat_lr | Acc Char | 3 | – |
| AGI | lsat_rc | Acc Char | 3 | – |
| AGI | sat_math | Acc Char | 5 | – |
| ARC | Challenge | Acc Char | 0 | – |
| ARC | Easy | Acc Char | 0 | – |
| BBH | boolean_expressions | Exact Match | 0 | – |
| BBH | causal_judgement | Exact Match | 0 | – |
| BBH | date_understanding | Exact Match | 0 | – |
| BBH | disambiguation_qa | Exact Match | 0 | – |
| BBH | dyck_languages | Exact Match | 0 | – |
| BBH | formal_fallacies | Exact Match | 0 | – |
| BBH | geometric_shapes | Exact Match | 0 | – |
| BBH | hyperbaton | Exact Match | 0 | – |
| BBH | logical_deduction_3_objects | Exact Match | 0 | – |
| BBH | logical_deduction_5_objects | Exact Match | 0 | – |
| BBH | logical_deduction_7_objects | Exact Match | 0 | – |
| BBH | movie_recommendation | Exact Match | 0 | – |
| BBH | multistep_arithmetic_two | Exact Match | 0 | – |
| BBH | navigate | Exact Match | 0 | – |
| BBH | object_counting | Exact Match | 0 | – |
| BBH | penguins_in_a_table | Exact Match | 0 | – |
| BBH | reasoning_about_colored_objects | Exact Match | 0 | – |
| BBH | ruin_names | Exact Match | 0 | – |
| BBH | salient_translation_error_detection | Exact Match | 0 | – |
| BBH | snarks | Exact Match | 0 | – |

**Table 1 – continued from previous page**

| Benchmark | Subset | Metric | Shots | Additional |
|---|---|---|---|---|
| BBH | sports_understanding | Exact Match | 0 | – |
| BBH | temporal_sequences | Exact Match | 0 | – |
| BBH | tracking_shuffled_objects_3 | Exact Match | 0 | – |
| BBH | tracking_shuffled_objects_5 | Exact Match | 0 | – |
| BBH | tracking_shuffled_objects_7 | Exact Match | 0 | – |
| BBH | web_of_lies | Exact Match | 0 | – |
| BBH | word_sorting | Exact Match | 0 | – |
| BoolQ | – | Acc Token | 0 | – |
| CommonSenseQA | – | Acc Char | 7 | – |
| COPA | – | Acc Char | ? | – |
| DROP | – | Exact Match | 0 | – |
| DROP | – | F1 | 0 | – |
| ETHOS | – | Acc Char | 0 | – |
| GSM8K | – | Exact Match | 0 | 100 Gen |
| GSM8K | – | F1 | 0 | 100 Gen |
| GSM8K | – | Exct Mtch maj1@100 | 0 | 100 Gen |
| GSM8K | – | F1 maj1@100 | 0 | 100 Gen |
| HellaSwag | – | Acc Char | ? | – |
| Human Eval | – | pass@1 | 0 | 1 Gen |
| Human Eval | – | pass@1 | 0 | 200 Gen |
| Human Eval | – | pass@100 | 0 | 200 Gen |
| Inverse Scaling | hindsight_neglect | Exact Match | 0 | – |
| Inverse Scaling | into_the_unknown | Exact Match | 0 | – |
| Inverse Scaling | memo_trap | Exact Match | 0 | – |
| Inverse Scaling | modus_tollens | Exact Match | 0 | – |
| Inverse Scaling | neqa | Exact Match | 0 | – |
| Inverse Scaling | pattern_matching_suppression | Exact Match | 0 | – |
| Inverse Scaling | redefine | Exact Match | 0 | – |
| Inverse Scaling | repetitive_algebra | Exact Match | 0 | – |
| Inverse Scaling | resisting_correction | Exact Match | 0 | – |
| Inverse Scaling | sig_figs | Exact Match | 0 | – |
| Kth Sentence | 128 | ROUGE-2 | 0 | – |
| Kth Sentence | 256 | ROUGE-2 | 0 | – |
| Kth Sentence | 512 | ROUGE-2 | 0 | – |
| Kth Sentence | 1024 | ROUGE-2 | 0 | – |
| Kth Sentence | 1536 | ROUGE-2 | 0 | – |
| Kth Sentence | 2048 | ROUGE-2 | 0 | – |
| Kth Sentence | 4096 | ROUGE-2 | 0 | – |
| Kth Sentence | 8192 | ROUGE-2 | 0 | – |
| MBPP | – | pass@1 | 3 | 80 Gen |
| MBPP | – | pass@80 | 3 | 80 Gen |
| MMLU | abstract_algebra | Acc Char | 5 | – |
| MMLU | anatomy | Acc Char | 5 | – |
| MMLU | astronomy | Acc Char | 5 | – |
| MMLU | business_ethics | Acc Char | 5 | – |
| MMLU | clinical_knowledge | Acc Char | 5 | – |
| MMLU | college_biology | Acc Char | 5 | – |
| MMLU | college_chemistry | Acc Char | 5 | – |
| MMLU | college_computer_science | Acc Char | 5 | – |
| MMLU | college_mathematics | Acc Char | 5 | – |
| MMLU | college_medicine | Acc Char | 5 | – |
| MMLU | college_physics | Acc Char | 5 | – |
| MMLU | computer_security | Acc Char | 5 | – |
| MMLU | conceptual_physics | Acc Char | 5 | – |
| MMLU | econometrics | Acc Char | 5 | – |
| MMLU | electrical_engineering | Acc Char | 5 | – |
| MMLU | elementary_mathematics | Acc Char | 5 | – |

**Table 1 – continued from previous page**

| Benchmark | Subset | Metric | Shots | Additional |
|---|---|---|---|---|
| MMLU | formal_logic | Acc Char | 5 | – |
| MMLU | global_facts | Acc Char | 5 | – |
| MMLU | high_school_biology | Acc Char | 5 | – |
| MMLU | high_school_chemistry | Acc Char | 5 | – |
| MMLU | high_school_computer_science | Acc Char | 5 | – |
| MMLU | high_school_european_history | Acc Char | 5 | – |
| MMLU | high_school_geography | Acc Char | 5 | – |
| MMLU | high_school_government_and_politics | Acc Char | 5 | – |
| MMLU | high_school_macroeconomics | Acc Char | 5 | – |
| MMLU | high_school_mathematics | Acc Char | 5 | – |
| MMLU | high_school_microeconomics | Acc Char | 5 | – |
| MMLU | high_school_physics | Acc Char | 5 | – |
| MMLU | high_school_psychology | Acc Char | 5 | – |
| MMLU | high_school_statistics | Acc Char | 5 | – |
| MMLU | high_school_us_history | Acc Char | 5 | – |
| MMLU | high_school_world_history | Acc Char | 5 | – |
| MMLU | human_aging | Acc Char | 5 | – |
| MMLU | human_sexuality | Acc Char | 5 | – |
| MMLU | international_law | Acc Char | 5 | – |
| MMLU | jurisprudence | Acc Char | 5 | – |
| MMLU | logical_fallacies | Acc Char | 5 | – |
| MMLU | machine_learning | Acc Char | 5 | – |
| MMLU | management | Acc Char | 5 | – |
| MMLU | marketing | Acc Char | 5 | – |
| MMLU | medical Genetics | Acc Char | 5 | – |
| MMLU | miscellaneous | Acc Char | 5 | – |
| MMLU | moral_disputes | Acc Char | 5 | – |
| MMLU | moral_scenarios | Acc Char | 5 | – |
| MMLU | nutrition | Acc Char | 5 | – |
| MMLU | philosophy | Acc Char | 5 | – |
| MMLU | prehistory | Acc Char | 5 | – |
| MMLU | professional_accounting | Acc Char | 5 | – |
| MMLU | professional_law | Acc Char | 5 | – |
| MMLU | professional_medicine | Acc Char | 5 | – |
| MMLU | professional_psychology | Acc Char | 5 | – |
| MMLU | public_relations | Acc Char | 5 | – |
| MMLU | security_studies | Acc Char | 5 | – |
| MMLU | sociology | Acc Char | 5 | – |
| MMLU | us_foreign_policy | Acc Char | 5 | – |
| MMLU | virology | Acc Char | 5 | – |
| MMLU | world_religions | Acc Char | 5 | – |
| NaturalQuestions | – | Exact Match | 0 | – |
| OpenBookQA | – | Acc Completion | 0 | – |
| PIQA | – | Acc Char | 0 | – |
| QuAC | – | F1 | 0 | – |
| RACE | High School | Acc Char | 0 | – |
| RACE | Middle School | Acc Char | 0 | – |
| SIQA | – | Acc Char | 0 | – |
| SQuAD | – | Exact Match | 0 | – |
| SciBench | atkins | Fuzzy Match | 0 | – |
| SciBench | calculus | Fuzzy Match | 0 | – |
| SciBench | chemmc | Fuzzy Match | 0 | – |
| SciBench | class | Fuzzy Match | 0 | – |
| SciBench | diff | Fuzzy Match | 0 | – |
| SciBench | fund | Fuzzy Match | 0 | – |
| SciBench | matter | Fuzzy Match | 0 | – |
| SciBench | quan | Fuzzy Match | 0 | – |

**Table 1 – continued from previous page**

| Benchmark | Subset | Metric | Shots | Additional |
|---|---|---|---|---|
| SciBench | stat | Fuzzy Match | 0 | – |
| SciBench | thermo | Fuzzy Match | 0 | – |
| TLDR | – | ROUGE-2 | 0 | – |
| TLDR | – | ROUGE-L | 0 | – |
| TriviaQA | – | Exact Match | 0 | – |
| WinoGrande | – | Acc Char | 0 | – |
| Xsum | – | ROUGE-2 | 1 | – |

A.2 DATA: HUMAN EVALUATION SCORES

Human data annotators were hired to evaluate outputs of chat language models (LMs) in single-turn and multi-turn conversations using a Likert scale (Likert, 1932) from 1 to 7. The conversations were constructed within our taxonomy of areas-categories-subcategories (Sec. 3; Fig. 2). Each conversation was evaluated by at least three unique humans for a combined total of 2104 unique human annotators. Our human annotators scored 11291 single-turn conversations and 2081 multi-turn conversations.

Table 2: **Human Evaluation Areas, Categories, and Subcategories.**

| Area | Category | Subcategory |
|---|---|---|
| Dialogue | Adversarial Dishonesty | Adversarial Dishonesty |
| Dialogue | Adversarial Harmfulness | Adversarial Harmfulness |
| Dialogue | Advice | Casual advice & recommendations |
| Dialogue | Advice | Personal & interpersonal relationships |
| Dialogue | Brainstorming | Brainstorming |
| Dialogue | Classification | Classification |
| Dialogue | Closed QA | Closed QA |
| Dialogue | Code | Code |
| Dialogue | Conversational / Entertainment | Conversational / Entertainment |
| Dialogue | Conversational/Entertainment | Conversational/Entertainment |
| Dialogue | Dialogue | Dialogue |
| Dialogue | Extraction | Extraction |
| Dialogue | Factual Questions | Factual Questions |
| Dialogue | Identity / Personas | Famous historical personalities |
| Dialogue | Identity / Personas | Fictional characters |
| Dialogue | Identity / Personas | No character (it's just an AI) |
| Dialogue | Identity / Personas | Public figures |
| Dialogue | Identity / Personas | Synthetic (made up) characters |
| Dialogue | Language Assistance | Language Assistance |
| Dialogue | Math | Math |
| Dialogue | Mathematical Reasoning | Mathematical Reasoning |
| Dialogue | Open QA | Open QA |
| Dialogue | Procedural Questions | Procedural Questions |
| Dialogue | Reasoning (math / problem-solving) | Reasoning (math / problem-solving) |
| Dialogue | Recommendations / Brainstorming | Recommendations / Brainstorming |
| Dialogue | Rewriting | Rewriting |
| Dialogue | Safety | Safety |
| Dialogue | Summarization | Summarization |
| Dialogue | Writing | Writing |
| Dialogue | Writing & Content Creation | Writing & Content Creation |
| Factual Questions | Cultural & social topics | Arts & literature |
| Factual Questions | Cultural & social topics | History & traditions |
| Factual Questions | Cultural & social topics | Popular culture & media |
| Factual Questions | Cultural & social topics | Religion & spirituality |
| Factual Questions | Cultural & social topics | Social issues & current events |
| Language assistance | Grammar, spelling, & vocabulary | English slang |

**Table 2 – continued from previous page**

| Area | Category | Subcategory |
|---|---|---|
| Language assistance | Grammar, spelling, & vocabulary | Grammar & syntax |
| Language assistance | Grammar, spelling, & vocabulary | Language conventions & style |
| Language assistance | Grammar, spelling, & vocabulary | Spelling & orthography |
| Language assistance | Grammar, spelling, & vocabulary | Vocabulary & word choice |
| Recommendations | Entertainment suggestions | Books, authors, & genres |
| Recommendations | Entertainment suggestions | Games, apps, & digital content |
| Recommendations | Entertainment suggestions | Movies, TV shows, & streaming content |
| Recommendations | Entertainment suggestions | Music, songs, & artists |
| Recommendations | Personal & professional development | Health, wellness, & self-improvement tips |
| Recommendations | Personal & professional development | Job search & career advice |
| Recommendations | Personal & professional development | Networking & mentorship opportunities |
| Recommendations | Personal & professional development | Skill-building resources & courses |
| Writing & content creation | Creative writing | Articles & reviews |
| Writing & content creation | Creative writing | Fictional stories & narrative |
| Writing & content creation | Creative writing | In-the-style-of writing |
| Writing & content creation | Creative writing | Poetry & songwriting |
| Writing & content creation | Creative writing | Social media posts |
| Writing & content creation | Summarization & editing | Content restructuring & organization |
| Writing & content creation | Summarization & editing | Proofreading |
| Writing & content creation | Summarization & editing | Style & tone adjustments |

## A.3 ANALYSES: LINEAR REGRESSION

Due to space limitations in the main text, we deferred citations regarding generalization of overparameterized models to here. For a nonexhaustive list, please see Vallet (1989); Krogh & Hertz (1991); Geman et al. (1992); Krogh & Hertz (1992); Opper (1995); Duin (2000); Spigler et al. (2018); Belkin et al. (2019); Bartlett et al. (2020); Belkin et al. (2020); Nakkiran et al. (2021); Poggio et al. (2019); Advani et al. (2020); Liang & Rakhlin (2020); Adlam & Pennington (2020); Rocks & Mehta (2022b; 2021; 2022a); Mei & Montanari (2022); Hastie et al. (2022); Bach (2023); Schaeffer et al. (2023a;b); Curth et al. (2024); Schaeffer et al. (2024b).

## B    STATISTICS OF HUMAN EVALUATIONS

As exploratory data analysis, we calculated and examined basic statistics of the human evaluations. Fig. 8 showcases how many turns (i.e., back and forth messages) are in each sample evaluated by human annotators (left) and how many human annotators evaluated each sample (right). Fig. 9 shows the empirical cumulative distributions functions of the average of human annotators' scores per datum (left) and the standard deviation of human annotators' scores per datum (right). Fig. 10 visualizes the joint distribution of means and standard deviations of human annotators' scores per datum as both a scatterplot (left) and a kernel density estimate (right).

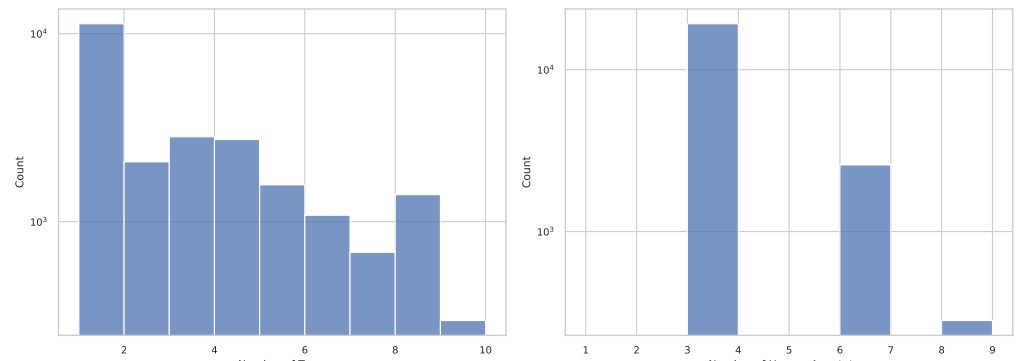

Figure 8: **Statistics of Human-Evaluated Data.** Left: Number of turns per datum. Right: Number of human annotators per human-evaluated datum.

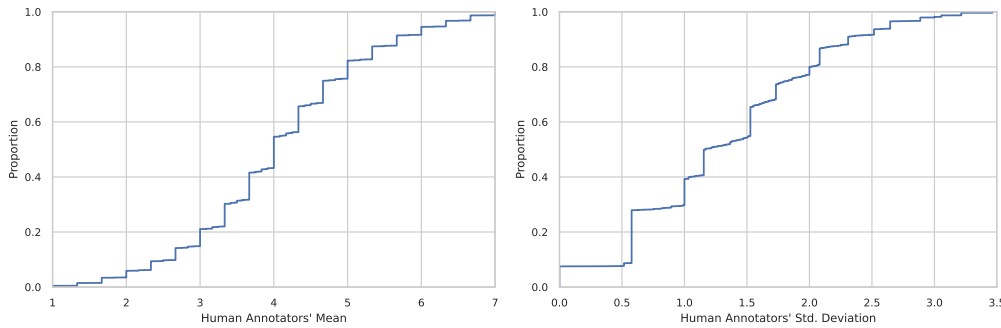

Figure 9: **Empirical Cumulative Distribution Functions of Human Annotators' Scores.** Left: Average of human annotators' score per annotated sample. Right: Human annotators' standard deviation per annotated sample.

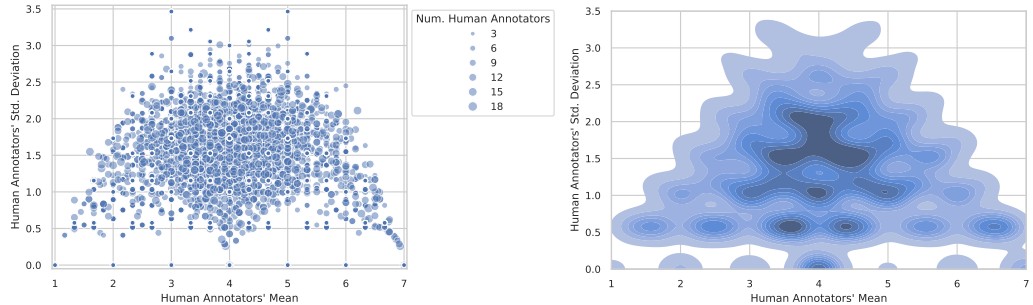

Figure 10: **Joint Distribution of Human Annotators' Average Scores per Datum vs Standard Deviation per Datum.** Left: Scatterplot. Right: Kernel Density Estimate.

# C   CORRELATION METRICS ARE THEMSELVES HIGHLY CORRELATED

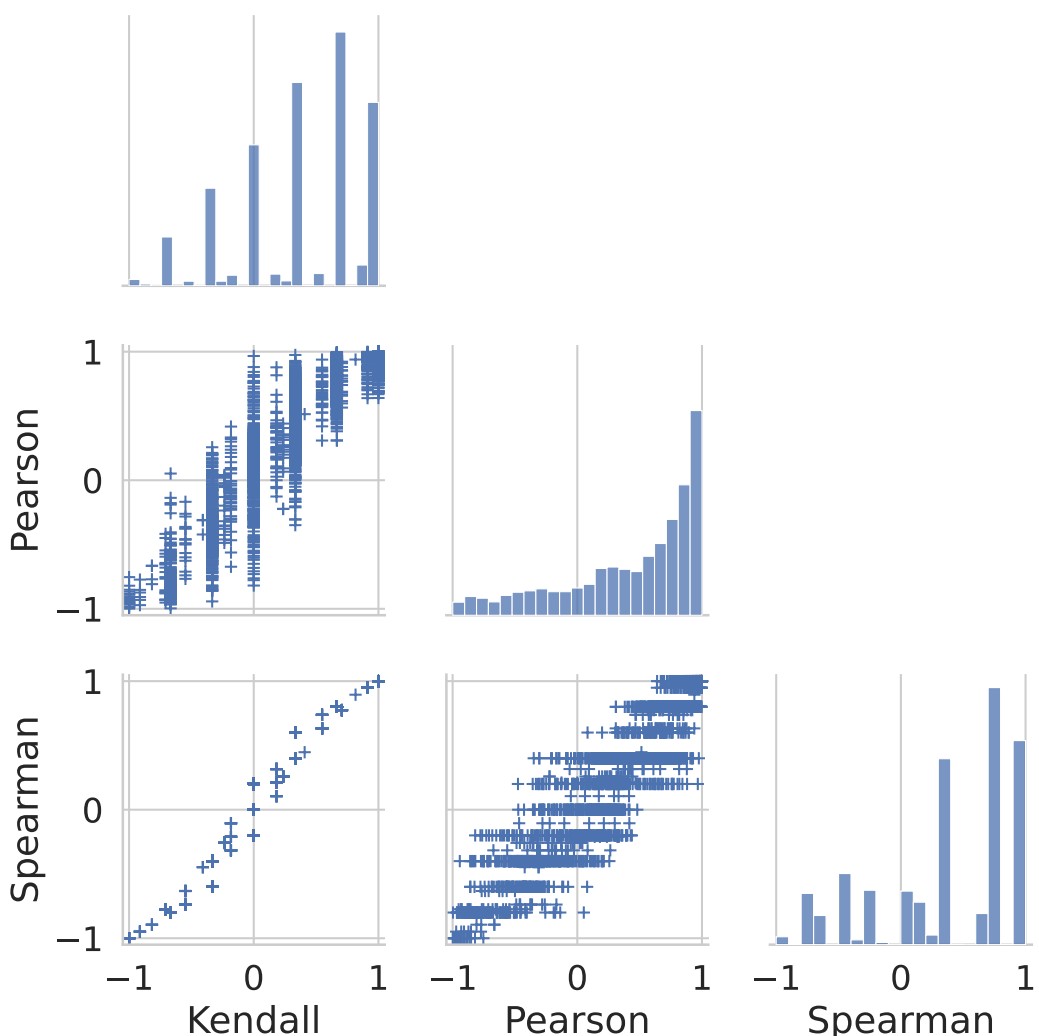

Figure 11: **Correlation of Academic-Human Evaluation Correlations Under Different Correlation Metrics**. For each pair of human evaluation (area and category) and NLP benchmark (benchmark and subset), we computed the correlation between scores under one of 3 correlation metrics: Pearson, Spearman and Kendall. We then looked at how correlated the correlation scores under the 3 correlation metrics are. In general, all 3 are correlation metrics yield correlated scores. This demonstrates that the choice of correlation metric is relatively less important.

## D CORRELATION MATRICES BETWEEN HUMAN EVALUATIONS AND NLP BENCHMARKS AND THEIR SINGULAR VALUE DECOMPOSITIONS

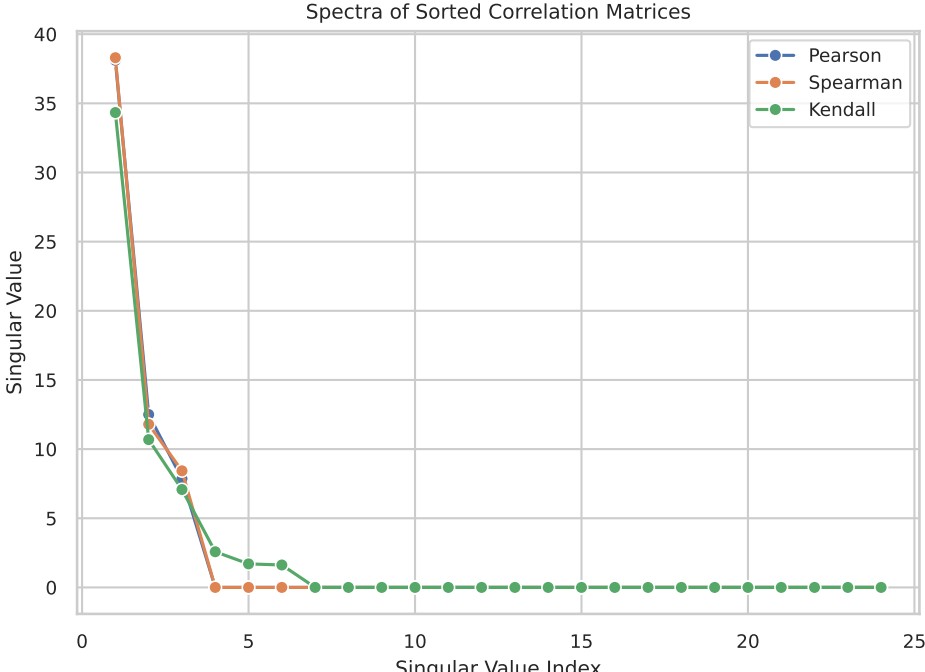

Figure 12: **Spectra of Human Evaluation-NLP Benchmark Correlation Matrices**. Because the correlation matrices are computed over four Chat Llama 2 models, the maximum matrix rank is 4. Howeer, both Pearson and Spearman correlation matrices have only 3 non-zero singular values.

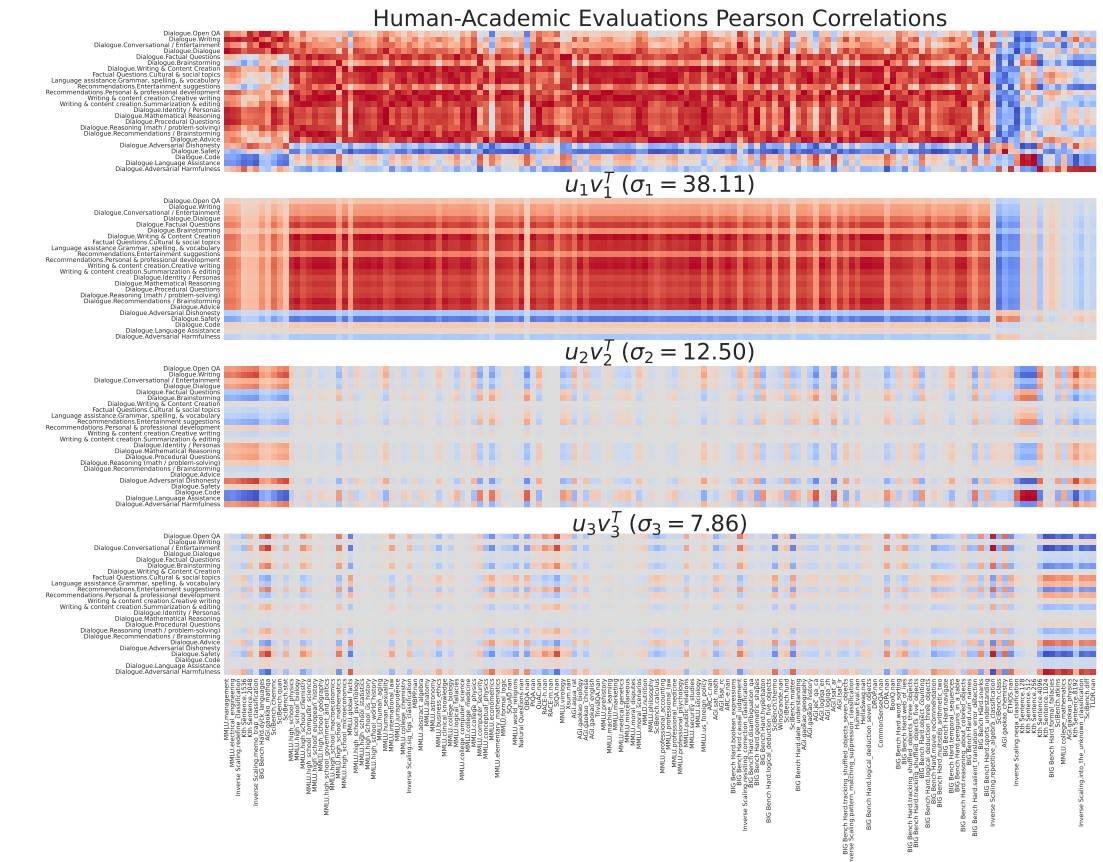

Figure 13: **Pearson Correlation Matrix between Human Evaluations and NLP Benchmarks.** The Pearson correlation matrix has 3 non-zero singular values, with corresponding modes shown in the last 3 rows.

The first component of the Pearson correlation matrix divides the human evaluations and NLP benchmarks into 3 groups (Fig. 6B): one group that is broadly uncorrelated, and two unequally-sized groups that are self-correlated and mutually anti-correlated. The *uncorrelated group* consists of human evaluations Dialogue:Code and Dialogue:Language Assistance, as well as NLP benchmarks Kth-sentence, TLDR, SciBench's Atkins and Differential Equations, MMLU's College Math and BBH's Formal Fallacies. The *smaller self-correlated group* consists of Dialogue:Adversarial Dishonesty and Safety:Harmlessness as well as ETHOS (a hate speech detection benchmark), Inverse Scaling NEQA (a negation question-answering benchmark) and AGI Gaokao Chemistry, whereas the *larger self-correlated group* consists of almost all other human evaluations and NLP benchmarks. This is more clearly visually displayed in the Spearman correlation matrix (App. Fig. 14).

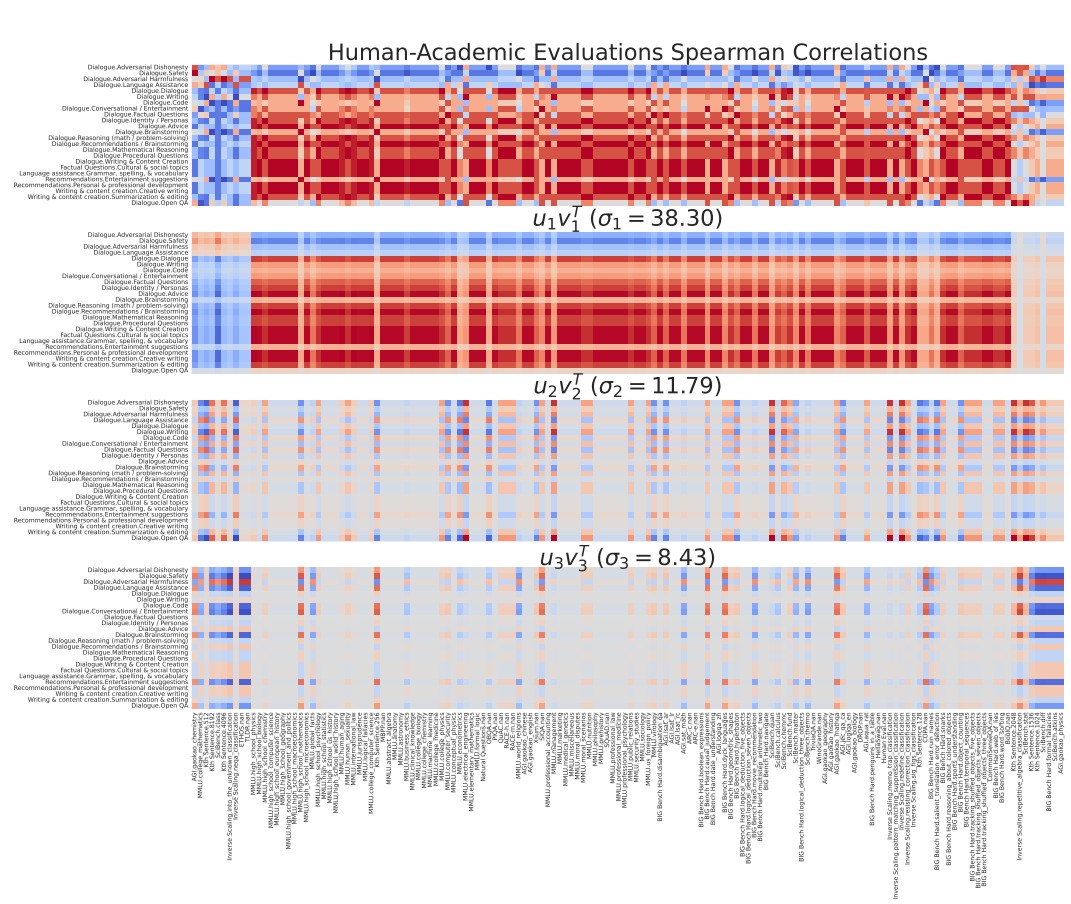

Figure 14: **Spearman Correlation Matrix between Human Evaluations and NLP Benchmarks.** The Spearman correlation matrix also has 3 non-zero singular values, with corresponding modes shown in the last 3 rows.

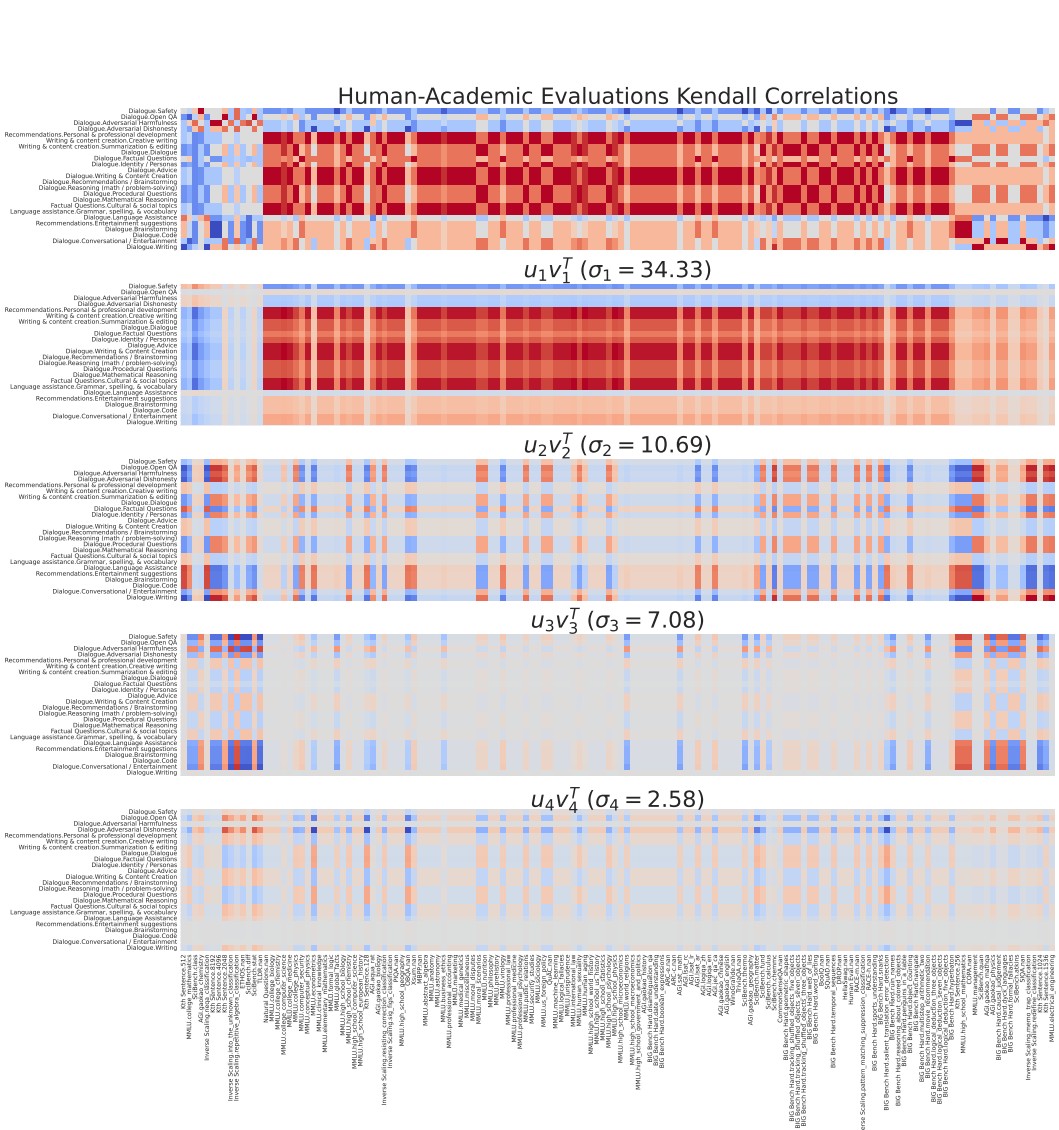

Figure 15: **Kendall Correlation Matrix between Human Evaluations and NLP Benchmarks.** The Kendall correlation matrix has four non-zero singular values, with corresponding modes shown in the last four rows.

# E EMPIRICAL SCALING BEHAVIOR OF HUMAN EVALUATIONS AND NLP BENCHMARKS

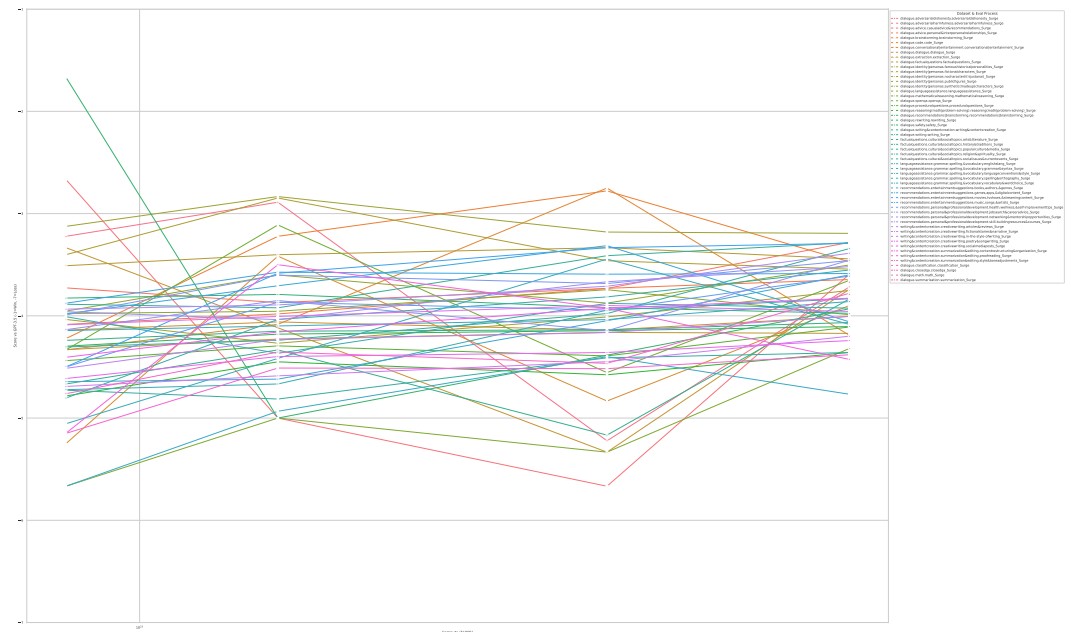

Figure 16: **Empirical Scaling Behavior of Human Evaluations with Increasing Compute.**

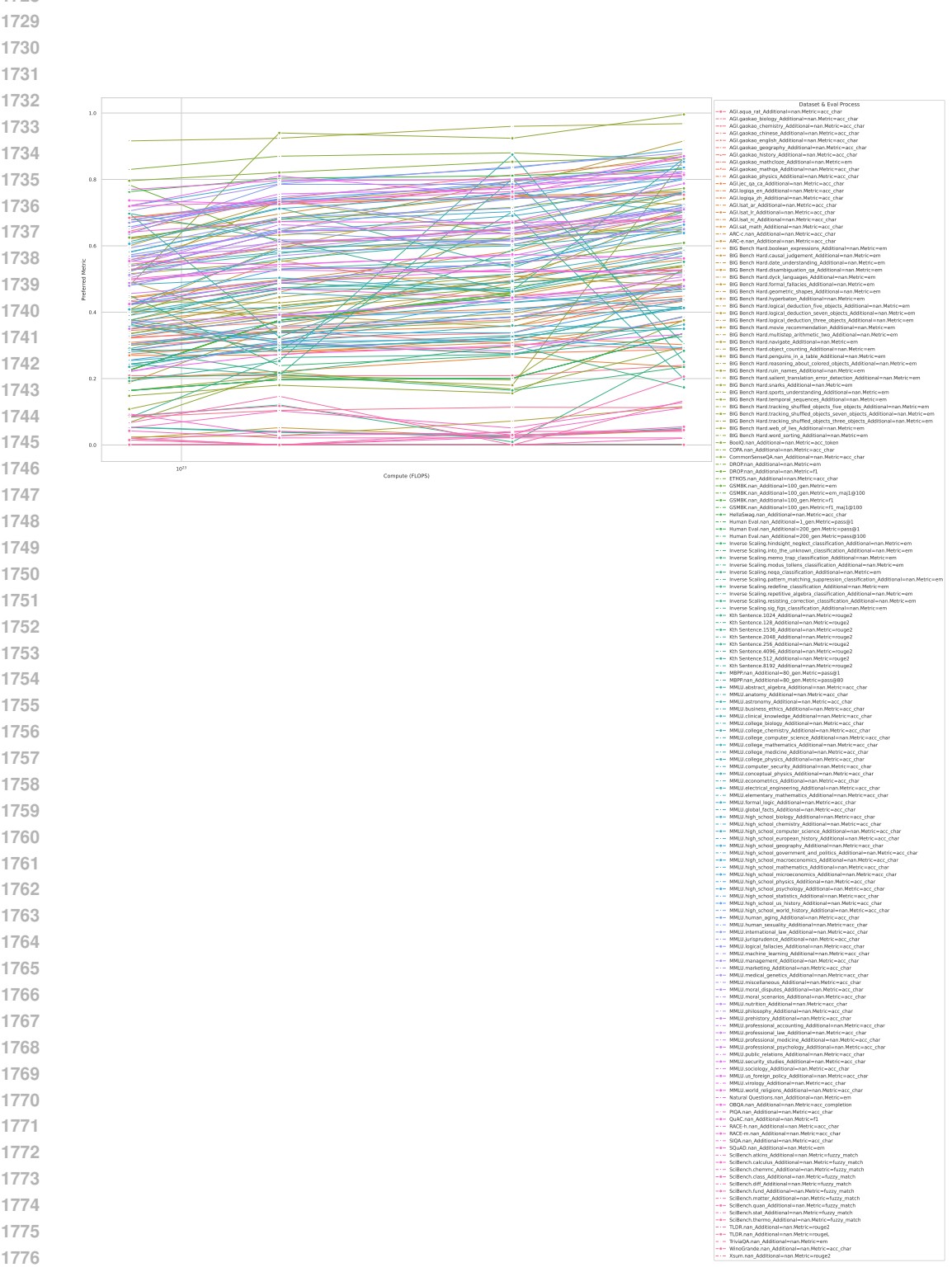

Figure 17: **Empirical Scaling Behavior of NLP Benchmarks with Increasing Compute.**

# F COEFFICIENTS OF LEAVE-ONE-OUT CROSS-VALIDATED LINEAR REGRESSIONS

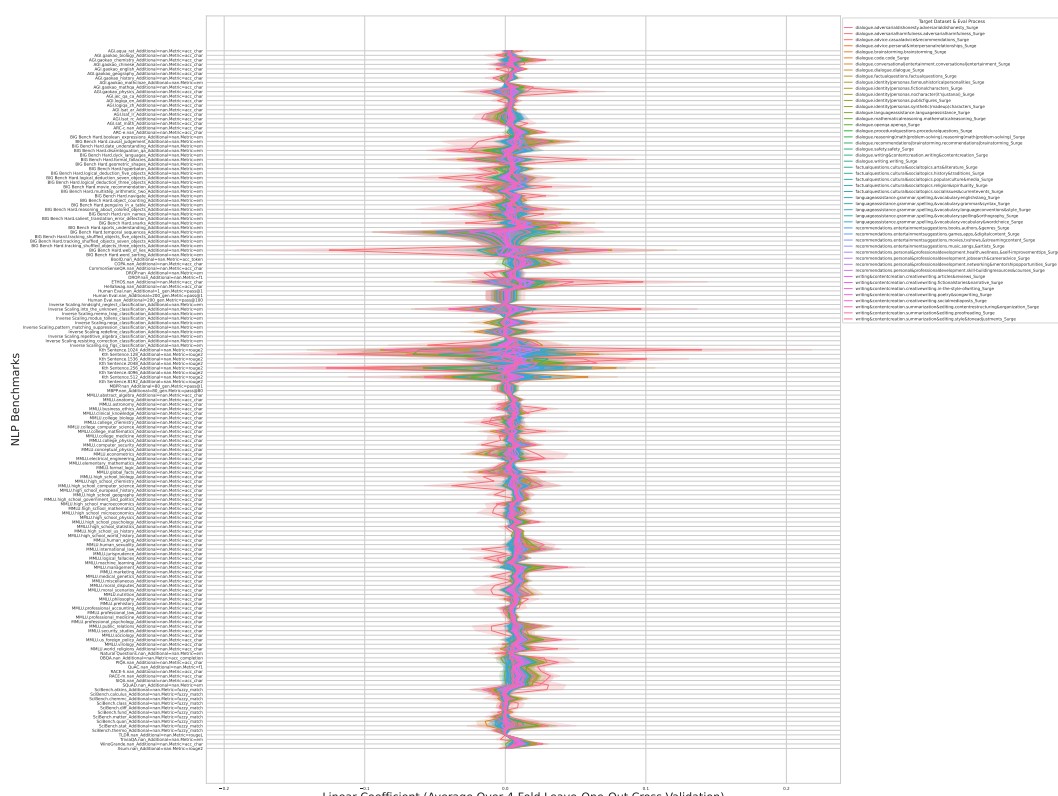

Figure 18: **Linear Coefficients of NLP Benchmarks in Predicting Human Evaluations.** For each human evaluation area, category and subcategory, we visualize the learned linear parameters per NLP benchmark averaged over the 4-fold leave-one-out cross validation process. For interpretation of results, see Sec. 5.

