# OpenReview forum: "Correlating and Predicting Human Evaluations of Language Models from Natural Language Processing Benchmarks"
_ICLR.cc/2025/Conference — Submitted to ICLR 2025_

### Official Review · Reviewer_Khx9 · 2024-10-29

**Soundness:** 2
**Presentation:** 1
**Contribution:** 1
**Rating:** 3
**Confidence:** 4

**Summary:**

This paper studies the relationship between NLP benchmarks and human evaluation results and aims to understand what roles NLP benchmarks should play in the era of LLM. They conduct human evaluations on four Llama 2 chat models and calculate the correlation between human evaluation results NLP benchmarks, spanning from open-domain QA, MMLU, and safety/adversarial datasets. They find that most NLP benchmarks correlate well with human evaluation results, and it is possible to predict human evaluation results based on scores on NLP benchmarks.

**Strengths:**

- This paper studies a very important problem: whether scores on NLP benchmarks correlate with human evaluation results. This can potentially guide researchers to construct better benchmarks
- This paper studies the possibility of using NLP benchmarks to predict human evaluation results. Considering the efforts of human evaluation, the problem studied in this paper can help us develop LLMs faster.

**Weaknesses:**

- The experiment parts are highly unclear and hard to comprehend. It is unclear how the correlations are calculated between human evaluation and NLP benchmark scores. There is even no **Experiment Setup** section in this paper, and the part that most looks like the experiment setting is the first seven lines of Section 3. After repeatedly reading those lines, I still cannot understand how the correlations are calculated. Precisely,
    - How do you aggregate the scores of different shots?
    - Why do you aggregate the results of different shots?
    - What is the number of shots?
    - How is the prompt formatted?
    - How are the demonstrations in the few-shot selected?
    - Where does the number *150* on Line 148 (page 3) come from?
    - How is the human evaluation conducted? How many samples are there in the single-turn and multi-turn dialogues? How are the topics selected? What is the distribution of the data?
    - If the paper only uses four models, is the correlation coefficient calculated using only the benchmark scores of 4 models and the human evaluation results of the models? This means we are only calculating the correlation coefficient between two sets for numbers with only four elements in each set.

- There are only four models used in this paper: the four chat models in Llama-2 with different numbers of parameters. The abilities of those models are very distinct, so it is easier for human evaluators or NLP benchmarks to distinguish the strengths of these models. A more challenging and realistic scenario is to consider more LLMs whose abilities are more diverse.

- The figures in the paper are terribly and poorly formatted. Those figures do not seem like they are designed to be read. The font sizes in the figures are too small to read and clustered together. I need to zoom in to 400% on my computer to see the words.

- Section 3.3 is highly unclear, without explaining what the *communities* this section is discussing and with no experiment settings that allow the readers to understand what is happening now.

Considering that the experiment setting is highly unclear and the results are poorly presented, it is impossible to evaluate the contribution of this work. The paper requires major refinement. However, the paper studies an important problem, and I encourage the authors to keep working on this topic.

**Questions:**

- Q1. How do the authors conduct the experiment using the Llama-2-30b model?  In fact, there is no 30b model in the LLama2 series, and I assume the authors are referring to the Llama-2-34b model. However, even Llama-2-34b-chat (or the base model) is not officially released, so I wonder how this paper conduct experiments using Llama-2-34b-chat.

---

> ### Author Response · Authors · 2024-11-26
> **Response to Reviewer Khx9 (Part 1)**
>
> Thank you for your review! We are grateful to hear you write that our paper studies a very important problem and that the insights of our paper could help design NLP benchmarks that are more predictive of human evaluations.
>
> We are working on improving the manuscript and should have a more complete version posted in 1-2 days.
>
> In the interim, to address a subset of the concerns you raised:
>
> > The experiment parts are highly unclear and hard to comprehend. It is unclear how the correlations are calculated between human evaluation and NLP benchmark scores. There is even no Experiment Setup section in this paper
>
> We are adding a detailed Experimental Methodology as Appendix A to describe our methodology in meticulous detail.
>
> As an overarching comment, all of our NLP benchmark scores are computed in the “default” or “standard” manner, e.g., as one would find in Meta’s Llama 2 paper, which is the work that our paper built on top of. Please see the new Appendix A.1; we will add more information about human evaluations today or tomorrow.
>
> > How do you aggregate the scores of different shots?
>
> > Why do you aggregate the results of different shots?
>
> To clarify, we do not aggregate scores of different shots.  Could you please point us towards where you read this? We’re unclear what in our manuscript gave you this misconception and we would like to correct whatever text gave this impression.
>
> To clarify, as you may know, different NLP benchmarks are oftentimes evaluated with different numbers of shots. For example, we evaluate AGI 5-shot, BoolQ 0-shot, CommonSenseQA 7-shot, etc. We use whatever number of shots is considered “standard” for each benchmark and do not experiment with these hyperparameters.
>
> > What is the number of shots?
>
> We have added Table 1 to Appendix A detailing the number of shots for each benchmark. As you may know, different benchmarks are evaluated with different numbers of examples. The number of shots for each benchmark were chosen to match the Llama 2 paper and we did not explore the effects of changing the number of shots.
>
> > How is the prompt formatted?
>
> > How are the demonstrations in the few-shot selected?
>
> Demonstrations and prompts are selected and formatted in the “standard” or “default” manner for each NLP benchmark. We followed the exact prompt selection and formatting from the Llama 2 paper.
>
> > Where does the number 150 on Line 148 (page 3) come from?
>
> ~150 on Line 148 is the number of NLP benchmark scores per model. The specific number is 160. This number arises because some NLP benchmarks have multiple subsets that we do not aggregate over. For instance, ARC has two subsets (ARC-Easy and ARC-Challenge), so each model receives two scores on the ARC benchmark. We stated this on Line 138: “Some of these benchmarks (e.g., MMLU) contain subsets (e.g., Jurisprudence) that we treat individually.” We would welcome language to help us better communicate this point.
>
> > How is the human evaluation conducted? How many samples are there in the single-turn and multi-turn dialogues? How are the topics selected? What is the distribution of the data?
>
> The human evaluations were conducted by contracting with a well-known data labeling company (redacted to preserve anonymity). The methodology for querying humans is described in lines 108 to 127. For single-turn evaluations, we have 1917 samples. Regarding how the topics were selected, as we stated on Line 123, “This taxonomy was chosen to broadly cover common use-cases of Chat LMs.”
>
> Regarding “What is the distribution of the data?”, could you please be more specific? What exactly would you like to know?
>
> > If the paper only uses four models, is the correlation coefficient calculated using only the benchmark scores of 4 models and the human evaluation results of the models? This means we are only calculating the correlation coefficient between two sets for numbers with only four elements in each set.
>
> Yes, this is correct. We stated this on line 149: “We then computed three standard notions of correlation over the 4 average scores per model.”
>
> > There are only four models used in this paper: the four chat models in Llama-2 with different numbers of parameters. The abilities of those models are very distinct, so it is easier for human evaluators or NLP benchmarks to distinguish the strengths of these models. A more challenging and realistic scenario is to consider more LLMs whose abilities are more diverse.
>
> This misunderstands the goal of our experimental design. The goal is to understand how human evaluation scores change as NLP benchmark scores change. Consequently, we want to see as much variance as possible because higher variance provides stronger signal for both correlating and predicting NLP benchmark and human evaluation scores. If we considered models with nearly identical scores, then this analysis would become much harder. By choosing models of different strengths, our analysis could be more robust.

---

> ### Author Response · Authors · 2024-11-26
> **Response to Reviewer Khx9 (Part 2)**
>
> > The figures in the paper are terribly and poorly formatted. Those figures do not seem like they are designed to be read. The font sizes in the figures are too small to read and clustered together. I need to zoom in to 400% on my computer to see the words.
>
> It is difficult to visualize the data because of how many signals (i.e. human evaluations and NLP benchmarks) we are plotting. We are working to improve our visualizations currently.
>
> > Section 3.3 is highly unclear, without explaining what the communities this section is discussing and with no experiment settings that allow the readers to understand what is happening now.
>
> The experimental setting is the same as before - nothing has changed. We are simply plotting and discussing the data.
>
> Regarding what “community” means, the term “community” is an (informal) reference to community detection in graphs (https://en.wikipedia.org/wiki/Community_structure). We will be updating this section in 1-2 days.
>
> > Q1. How do the authors conduct the experiment using the Llama-2-30b model? In fact, there is no 30b model in the LLama2 series, and I assume the authors are referring to the Llama-2-34b model.
>
> Your assumption is correct. We will rename Llama-2-30B to Llama-2-34B.
>
> > However, even Llama-2-34b-chat (or the base model) is not officially released, so I wonder how this paper conduct experiments using Llama-2-34b-chat.
>
> Llama-2-34B has indeed not been publicly released. We asked for and received permission from Meta to use the model for the purposes of our study.
>
> **Thank you for your patience while we work to integrate your feedback to improve the manuscript**

---

> ### Author Response · Authors · 2024-12-03
> **Polite Nudge to Reviewer Khx9 to Please Respond**
>
> Dear Reviewer Khx9,
>
> We significantly revamped our manuscript and wrote a lengthy response to your review. We strongly believe your score of 1 is unjustified. Could we ask you to please respond?
>
> Thank you!

---

> > ### Comment · Reviewer_Khx9 · 2024-12-03
> > **Response to Author Responses**
> >
> > Thank you for your responses.
> > However, I do not think the responses and modifications increase the quality of the manuscript to a level that can be accepted. I appreciate the authors trying to clarify where my misunderstandings are from. However, I cannot directly point out where in the manuscript gives me such misunderstanding as the original version of the paper really did not include too many details for me to understand, so I can only guess. It is not about what the paper writes but instead what the paper didn't write.
> >
> > Given that the experiment settings are somewhat clearer in the revised version, I can increase my score to 3. However, **fatal weaknesses** remain in the paper, including (1) only using four Llama models whose strengths vary a lot; this is a very limited study to draw the conclusion that **benchmarks correlate well with human evaluation**. While the authors say this is a misunderstanding of their work, I don't believe this is a misunderstanding, as the abstract claims that *"benchmarks are broadly highly correlated with human evaluations"*. I want to point out that drawing a conclusion on only these four models is not convincing enough. This is also pointed out by another reviewer. (2) The figures are still not readable. I do not think such formatting is friendly to readers, making me doubt whether this paper is suitable for publishing. (3) I still do not understand what the term "community" refers to in Section 4.3. After reading the revision and rebuttal, I cannot fully understand what kind of analysis is used here. Again, there are no details on this.

---

> > > ### Comment · Reviewer_Khx9 · 2024-12-03
> > > **Response to Author's Nudge**
> > >
> > > Additionally, I would like to express my **strong objection** to the authors saying ***your score of 1 is unjustified***.
> > > In my original review, I justified my score using more than 350 words with clearly formatted bullet points. This is a number twice to four times what other reviewers have written in the weakness part. Of course, more words do not translate to a better review, but I am just saying that the amount of weaknesses I deem this paper to have is very significant. Those questions are what readers will ask when reading this paper, and the weaknesses I raised will surely be spotted by other readers. My review highlights the fatal weaknesses of the paper, spanning from **experiment soundness, significance and impact of the results, and presentation**. I provide questions that highlight why the paper is unclear and actually point out some places that are imprecise in the manuscript, which the authors acknowledge in their responses (e.g., wrong model size for Llama-2-34b, the number 150 in the paper). Those imprecise numbers create difficulty when a reader wants to reproduce the paper, and the job of the reviewers is to point them out. I believe this is what my review has done. All these weaknesses together justify the score I initially gave.

---

> ### Author Response · Authors · 2024-12-03
> **Response to Reviewer Khx9**
>
> Thank you for your detailed feedback and for increasing the score based on our clarifications. We apologize if our previous message came across as dismissive of your thorough review - that wasn't our intent. We also do appreciate your engaging; as we're sure you're aware, many ICLR reviewers aren't engaging.
>
> However, we feel we must respectfully address several points from your initial review that may have contributed to an unnecessarily low initial score:
>
> - Your review suggested we were aggregating results across different numbers of shots, which is incorrect.
> - All of our evaluations are standard and exactly follow prior work. Your objections about lack of clarity for how many shots were used, how prompts were formatted, etc. does not seem well founded.
> - You identified a lack of an "Experiment Setup" section as a serious shortcoming. While we agree that our manuscript can improve with additional clarification and we added Appendix A, our analyses are 3 simple analyses of two matrices (the human evaluation scores and the NLP benchmark scores): (i) correlations, (ii) singular value decomposition and (iii) linear regressions. Such simple analyses should not require significant explanation.
> - You questioned our access to Llama-2-34B as if some nefarious plot was afoot. Rather, we had explicit permission from Meta to use this model. This access actually strengthens our paper by providing evaluation data on a model not widely available to the community.
> - Moreover, your initial score of "1" is, in general, extremely harsh. Independent of our paper, we feel that "1"s should be given out extremely rarely and only for work that is exceptionally and egregiously inadequate.
>
> While the ICLR manuscript resubmission deadline has passed, we want to address your remaining concerns:
>
> - Regarding the small model set: While four models may seem limited, they provide a controlled experiment across model scale while holding architecture and training constant. This allows us to isolate how performance changes with scale.
>
> - Figure readability: We made a serious effort to (i) improve our figures, (ii) add additional figures and (iii) remove unhelpful figures. Rather than criticizing the figures, you could be significantly more helpful if you tell us how to improve our figures.
>
> - Community detection methodology: We will add an appendix section to Appendix A detailing this methodology. To explain here, community detection is a standard approach in network science. In our context, we have a bipartite graph, where the two node sets are human evaluations and NLP benchmarks, and the edge weights are the correlations between nodes. To identify such communities, we turn to the most basic linear algebra primitive (i.e. SVD) and study the different singular modes.
>
> Would you be open to providing additional feedback about how to improve further? Your critique has helped us identify important ways to make this work more rigorous and accessible, while still maintaining its core contributions.

---

> ### Comment · Reviewer_Khx9 · 2024-12-03
> **Responses to Authors' Responses (1/2)**
>
> I want to thank the authors again for their responses. I understand the concerns they have raised and appreciate the effort they have put into addressing the review. I also understand the frustration that can accompany receiving a low score. I welcome this opportunity to discuss the work further, as it allows me to clarify potential misunderstandings and ensure that readers have a complete perspective on how the paper was evaluated.
>
> However, I must address what appears to be a suggestion that the review was conducted without properly reading the paper. This suggestion is entirely unfounded, as I have thoroughly analyzed the submission. While the authors may not be making such a direct accusation, their responses create a context where this perception might arise. To avoid any confusion, I believe that additional clarification is necessary and will benefit the overall understanding of the evaluation process. Constructive discussions like this are valuable, and I remain committed to engaging in them professionally.
>
> > About whether the paper is aggregating the results of different shot. (Which it did not)
>
> Why do I have such a guess (not misunderstanding)? This is because, in the original version, the descriptions of the number of shots in the main content are simply "*we used standard evaluation processes for all academic benchmarks including prompt formatting, metrics, 0-shot/few-shot, etc.*" without further explanations. How can a reader know this means *"we use different numbers of shots and prompts for each dataset, following the previous work xxx"*? The original sentence seems to say each dataset uses multiple shots, so I can only guess those results are aggregated. Yes, my guess is incorrect, but **this is because the paper did not say anything about it**. This is clear evidence of insufficient experiment details. Next, just saying standard evaluation in NLP does not reveal any details for reproduction, not to mention that the original paper did not explicitly say, "We follow all the experiment settings in Llama-2". (If the paper did say this in the original version, please tell me and I am willing to apologize for that.) Last, the original version of the paper did not even have a dedicated Appendix section for the experiment setting and was only added on the last day of the author response period based on the reviewer's request. Based on all of these, this makes me believe that **the experiments in the initial version of the paper are not sound at all**. Now that the appendix sections have been added, I increase the score to 3 for this reason.
>
> > All of our evaluations are standard and exactly follow prior work. Your objections about lack of clarity for how many shots were used, how prompts were formatted, etc. does not seem well founded.
>
> The paper did not even **cite** the prior work they are mentioning, and the lack of clarity is, of course, well founded. Moreover, I do not believe there exists such as standard that says *dataset A uses $K$ shot and use prompt format xxx*. The author's statement on the existence of such a standard is highly questionable and not well-founded.
>
> > You identified a lack of an "Experiment Setup" section as a serious shortcoming. While we agree that our manuscript can improve with additional clarification and we added Appendix A, our analyses are 3 simple analyses of two matrices (the human evaluation scores and the NLP benchmark scores): (i) correlations, (ii) singular value decomposition and (iii) linear regressions. Such simple analyses should not require significant explanation.
>
> I want to stress it again: if the paper does not have an experiment setup section, no one can reproduce the results with high precision. No matter how simple they may seem, they are required. Reproducibility is the core of our discipline, and as a researcher working on evaluation, I know how painful it is to reproduce a paper without proper experiment setting details.
>
> > You questioned our access to Llama-2-34B as if some nefarious plot was afoot. Rather, we had explicit permission from Meta to use this model. This access actually strengthens our paper by providing evaluation data on a model not widely available to the community.
>
> **This is a strong accusation to say I am suggesting some nefarious plot was afoot**. My original question is *However, even Llama-2-34b-chat (or the base model) is not officially released, so I wonder how this paper conducts experiments using Llama-2-34b-chat.* How can one infer that I am questioning the integrity of how the authors access the model? I was simply questioning where and how one can get such a model, or did the authors train such a model by themselves? I think this question does not have such negative implications. Moreover, in the original paper, the number of parameters of the model is even wrong, making me more curious about how the model is obtained.

---

> > ### Comment · Reviewer_Khx9 · 2024-12-03
> > **Responses to Authors' Responses (2/2)**
> >
> > > Moreover, your initial score of "1" is, in general, extremely aggressive.
> >
> > I believe the score a paper deserves is highly subjective, and I already have been very very objective about this. I have listed reasons why the papers should be rejected. When adding them together, I think giving a score of 3 is not enough since it seems unfair for other papers that receive a 3, so I have no choice but to give this paper a 1. This paper does not have sound experiments, does not have a reasonable presentation, and does not have sufficient experiments to justify its main claim. Again, I feel unnecessary to argue whether this paper deserves a score of 1 since this is well and repeatedly justified in my review and responses.
> >
> > > Regarding the small model set: While four models may seem limited, they provide a controlled experiment across model scale while holding architecture and training constant. This allows us to isolate how performance changes with scale.
> >
> > Nothing is resolved since I simply believe using only four models is not sufficient, agreeing with evg9.
> >
> > > Figure readability: We made a serious effort to (i) improve our figures, (ii) add additional figures and (iii) remove unhelpful figures. Rather than criticizing the figures, you could be significantly more helpful if you tell us how to improve our figures.
> >
> > This response seems quite defensive and somewhat unpleasant for me when reading it. I also asked GPT-4o if this is polite, and this is what GPT-4o responded: *Your initial response might be seen as defensive because it indirectly suggests that the reviewer's critique of the figures might not be constructive ("Rather than criticizing the figures, you could be significantly more helpful if you tell us how to improve our figures"). This wording implies that the criticism wasn't helpful, which could be interpreted as a dismissal of the reviewer’s feedback.*  So, back to the problem. The issues about the figures are still not fixed in the latest revision. Please answer the following question: Can a reader read the tiny words in Figures 5 and 6 when the paper is printed on an A4 paper? I cannot. My recommendation? Make the font larger.
> >
> > > Community detection methodology: We will add an appendix section to Appendix A detailing this methodology. To explain here, community detection is a standard approach in network science. In our context, we have a bipartite graph, where the two node sets are human evaluations and NLP benchmarks, and the edge weights are the correlations between nodes. To identify such communities, we turn to the most basic linear algebra primitive (i.e. SVD) and study the different singular modes.
> >
> > This is another evidence that the paper is unclear. The paper, even in its revision, does not mention the term *cimmunity detection*. It is unlikely for a reader to know the community that suddenly appears here refers to that community in community detection. While I can guess what this means, and I know community detection, I don't think a good paper should make the readers guess what the paper is trying to say.

---

### Official Review · Reviewer_MpHR · 2024-11-03

**Soundness:** 4
**Presentation:** 2
**Contribution:** 3
**Rating:** 5
**Confidence:** 3

**Summary:**

This work attempts to explore the correlation or consistency between common NLP automatic evaluation benchmarks and human evaluations in analyzing and comparing the capabilities of language models. They cover a wide range of datasets and conduct experiments on four different sizes of Llama 2 models and GPT-3.5, employing human annotators to provide evaluation data. They find that there is a high correlation between automatic benchmarks and human evaluations, and they identify which benchmarks show stronger correlations. Furthermore, they also fit models to predict human evaluation scores of language models from academic evaluation scores.

**Strengths:**

The motivation and research questions of this work are very interesting and significant. Considering that language models are becoming increasingly powerful, many traditional NLP benchmarks may have lost their discriminative power, leading researchers to turn to human evaluations, which are more costly and harder to reproduce. By analyzing the consistency between NLP automatic evaluation benchmarks and human evaluations, this work aims to identify highly consistent benchmarks to simulate human evaluations, thereby reducing evaluation costs. Their experiments cover a large range of datasets and settings, including constructed various categories of human evaluation data and many common NLP automatic evaluation benchmarks, demonstrating a very comprehensive effort.

**Weaknesses:**

Although the research topic of this work is meaningful, it is also actually very complicated and corresponds to a more challenging analysis process. Even though the work has tried to handle the experimental data and present corresponding results as macroscopically as possible, their experimental analyses remain confusing and fail to help readers capture the main points. From Figure 1 onward, the clarity and readability of the charts decline rapidly, and by Figure 6, it becomes nearly impossible to extract any information as the fonts are extremely small and the visualized results are poorly presented.

Some analytical settings in the paper are unclear or somewhat unreasonable. For example, in line 149, what does the "evaluation process" refer to, and why are approximately 150 combinations calculated in total? What do they represent? Additionally, if I understand correctly, it seems unfair to compare human evaluation results across mixed task types with different NLP automatic evaluation benchmarks that may focus on testing certain different abilities.

**Questions:**

Please refer to Weaknesses.

---

> ### Author Response · Authors · 2024-12-03
> **Response to Reviewer MpHR**
>
> Thank you for your review. We appreciate that you felt the research topic of this work is meaningful. To address the concerns you raised:
>
> > their experimental analyses remain confusing and fail to help readers capture the main points.
>
> > From Figure 1 onward, the clarity and readability of the charts decline rapidly, and by Figure 6, it becomes nearly impossible to extract any information as the fonts are extremely small and the visualized results are poorly presented.
>
> This is a valid criticism that we significantly improved in our revised and resubmitted manuscript. To highlight key changes:
>
> - We better described our analyses in the main text (Section 3)
> - In case that was inadequate, we also created a new Appendix A detailing our experimental methodology including both data and analyses
> - We added Appendix B with basic analyses of our data to provide additional information to readers
> - We created new visualizations (specifically Figures 3, 5 and 7) that hopefully are more easily read and understood
> - We removed old visualizations that you and other reviewers felt did not add value and perhaps even subtracted value
> - We also cleaned up Figure 6 (but embarrassingly forgot to include it in our final submitted manuscript) to make the text more legible
>
> > For example, in line 149, what does the "evaluation process" refer to
>
> An evaluation process is the benchmark (e.g., MMLU), a possible subset (e.g., College Mathematics), plus any additional information necessary to specify how models are scored such as: the metric (accuracy, ROUGE-2, ROUGE-L, pass@k, f1 etc.), 0-shot or few-shot (and if few-shot, how many shots), whether answers are sampled from the model (and if so, how many), whether Chain-of-Thought reasoning is used, etc. We have clarified this in the main text and in our next Appendix A.
>
> > why are approximately 150 combinations calculated in total?
>
> ~150 on Line 148 is the number of NLP benchmark scores per model. The specific number is 160. This number arises because some NLP benchmarks have multiple subsets that we do not aggregate over. For instance, ARC has two subsets (ARC-Easy and ARC-Challenge), so each model receives two scores on the ARC benchmark. We stated this on Line 138: “Some of these benchmarks (e.g., MMLU) contain subsets (e.g., Jurisprudence) that we treat individually.”
>
> We clarified this in the main text and added Table 1 in Appendix A.1 to state exactly what benchmarks, subsets, metrics, few shot and generations we use.
>
> We would welcome language to help us better communicate this point.
>
> > Additionally, if I understand correctly, it seems unfair to compare human evaluation results across mixed task types with different NLP automatic evaluation benchmarks that may focus on testing certain different abilities.
>
> We added a paragraph to explain that this was an intentional decision and to motivate why:
>
> In this work, our aim was specifically to identify which NLP benchmark scores are predictive of human preferences on open-ended prompts representative of real-world chat model usage. We chose this approach to maximize the ecological validity and generalizability of the findings to real-world use cases. For a concrete example, we may want our chat language models (LMs) to excel at providing bespoke career advice; which NLP benchmarks provide useful signals for whether models are improving at such tasks?

---

### Official Review · Reviewer_evg9 · 2024-11-04

**Soundness:** 2
**Presentation:** 1
**Contribution:** 2
**Rating:** 3
**Confidence:** 3

**Summary:**

The paper studies the relationships between the evaluation results of automated NLP benchmarks and those of human evaluation. It mainly revolves around two research questions: how well human evaluations and NLP benchmarks are correlated with each other, how well NLP benchmarks can predict human evaluation. Specifically, the authors develop a set of 1917 prompts organized by areas, categories, and subcategories, selects four LLMs from Llama 2 family, gets their reponses to the prompts, and conducts a large-scale pairwise human evaluation. The evaluation results of the four models on many automated NLP benchmarks are also derived. Then, the paper analyzes the correlations between human evaluation and automated NLP benchmarks and finds that they are highly correlated in most cases. Furthermore, the authors decompose the correlation matrix into rank-one components and demonstrate the communities between human evaluations and NLP benchmarks. Finally, the authors tried to fit a regression model to predict the human evaluations with automatic evaluation results as inputs.

**Strengths:**

- The research question of this paper—the relationship between evaluation results from automated NLP benchmarks and human evaluations —is generally important and meaningful. Recently, numerous automated benchmarks and human evaluations have emerged separately, but there has been little research on the relationship between them.
- This paper covers many automated NLP benchmarks and includes a large-scale human evaluation, which lends a certain level of generality to its results.

**Weaknesses:**

Although the idea of this paper is beneficial, many obvious flaws diminish its value.

- This study uses only four LLMs, which is too few. This leads to
  - The correlations between automated NLP benchmarks and human evaluation are calculated merely from two four-dimension vectors, which is unreliable
  - Insufficient experiments for predicting human evaluation from automated NLP benchmarks, despite cross-validation conducted in the paper

- The paper lacks key details, including but not limited to how the prompt set used in human evaluation is obtained, the human evaluation process and its reliability (e.g. inter-annotator agreements), details of how the correlation is calculated (what is ~150 evaluation process?), the settings for linear regression. This not only creates difficulty in understanding but also raises doubt about the rigor of this study.

- The presentation of the paper could be improved. For instance, the font sizes in Fig 3, the upper part of Fig 4, and Fig 6 are too small, making it hard to read.

**Questions:**

- More LLMs should be covered in this study. I understand the computational cost during inference and the cost in human evaluation, but four LLMs are definitely too few to support subsequent experiments.
- I do need more details of the human evaluation in your study. What makes me most confused is the selection of the prompts. Why don't you use the same question sets as those of automated NLP benchmarks? If there are too many, you can sample from each dataset. Now there is a mismatch between the prompts (questions) in human evaluation and automated NLP benchmarks and the mapping relationship is not clear. Even if ignoring the mismatch issue, you should provide the number of prompts per area and categories used in human evaluation.
- The experiments of one-rank decomposition in Section 3.3 need to be further explained. Can you better state your motivation of conducting this decomposition and what insights can we draw from that?

---

> ### Author Response · Authors · 2024-11-26
> **Response to Reviewer evg9 (Part 1)**
>
> Thank you for your insightful review! We greatly appreciate you taking the time to thoroughly assess our work. We're glad you found the research question important and meaningful, and that you recognize the value in the generality and scale of both our NLP and human evaluations.
>
> To address the concerns you raised:
>
> > This study uses only four LLMs, which is too few
>
> > More LLMs should be covered in this study. I understand the computational cost during inference and the cost in human evaluation, but four LLMs are definitely too few to support subsequent experiments.
>
> We fully agree that testing more models would provide greater insight and robustness to the findings. Due to the high cost of collecting human evaluations (approximately $250k USD per model), we were constrained in the number of models we could include in this study. Please note that we did make extensive efforts to expand the models tested, but ran into data collection errors. Ultimately, we chose to focus on the 4 models from the Llama 2 family to maintain consistency in model architecture while still spanning a wide range of model scales. Your point is well-taken and we will emphasize this as a key limitation in our discussion section.
>
> > The correlations between automated NLP benchmarks and human evaluation are calculated merely from two four-dimension vectors, which is unreliable
>
> This is a fair concern given the small sample size. To increase the reliability of the correlations, what would you advise?
>
> To suggest one possibility, we could add confidence intervals estimated via bootstrapping. Specifically, we could resample the 4 models with replacement many times, compute the correlation on each sample, and report the 2.5 to 97.5 percentile range as 95% confidence intervals. We will also compute p-values via a permutation test to quantify the probability of observing correlations as extreme as we did under a null hypothesis of no correlation.
>
> Would these additional analyses be sufficient to address the sample size limitation? We welcome any other suggestions you may have.
>
> > Insufficient experiments for predicting human evaluation from automated NLP benchmarks, despite cross-validation conducted in the paper
>
> Similar to our response above, (1) why do you find leave-one-out cross validation insufficient, and (2) what analysis (or analyses) would you recommend?
>
> > The paper lacks key details, including but not limited to how the prompt set used in human evaluation is obtained, the human evaluation process and its reliability (e.g. inter-annotator agreements), details of how the correlation is calculated (what is ~150 evaluation process?), the settings for linear regression.
>
> We are adding a detailed Experimental Methodology to the Appendix to describe our methodology in meticulous detail. As an overarching comment, all of our NLP benchmark scores are computed in the “default” or “standard” manner, e.g., as one would find in Meta’s Llama 2 paper, which is the work that our paper built on top of. Please see the new Appendix A.1; we will add more information about human evaluations today or tomorrow.
>
> An “evaluation process” is the terminology we use to describe whatever additional information is necessary to describe how scores are computed on a dataset. An evaluation process is the metric, whether samples were generated (and if so, how many), whether 0-shot or few-shot prompting was used (and if so, how many exemplars), whether chain-of-thought prompting was used, etc. If you would advise different terminology to help us encapsulate all of these details, please let us know.

---

> > ### Comment · Reviewer_evg9 · 2024-11-29
> >
> > Thanks for your response. I still suggest you add more LLMs in addition to the Llama 2 family. I understand the cost of human evaluation, but you should plan carefully before conducting human evaluation: increasing the number of systems (LLMs) and decreasing the number of instances per system accordingly. Considering what connects the NLP benchmarks and the human evaluation in your study is only the four systems, it is hard to do other analyses.

---

> ### Author Response · Authors · 2024-11-26
> **Response to Reviewer evg9 (Part 2)**
>
> > I do need more details of the human evaluation in your study. What makes me most confused is the selection of the prompts. Why don't you use the same question sets as those of automated NLP benchmarks?
>
> You raise an interesting point about directly comparing human and automated evaluations on the same set of prompts. In this work, our aim was specifically to identify which NLP benchmark scores are predictive of human preferences on open-ended prompts representative of real-world chat model usage. We chose this approach to maximize the ecological validity and generalizability of the findings to practical applications. For a concrete example, we may want our chat LMs to excel at roleplaying as different characters or at building novel fantastical worlds; the question we want to know the answer to is: which NLP benchmarks provide useful signals on whether models are improving at such tasks?
>
> However, your suggested approach of comparing evaluations on the same prompts would provide valuable insight into the agreement between human and automated scores in a more controlled setting. We will note this as an important direction for future work in our discussion section. Thank you for the thought-provoking suggestion!
>
> > The presentation of the paper could be improved. For instance, the font sizes in Fig 3, the upper part of Fig 4, and Fig 6 are too small, making it hard to read.
>
> Thank you for alerting us to these readability issues in our figures. We are currently revising the identified figures to increase font sizes and ensure all text is clearly legible. The updated figures will be included in our next revision.
>
> **Thank you for your patience while we work to integrate your feedback to improve the manuscript.**

---

> ### Author Response · Authors · 2024-12-03
> **Response to Reviewer evg9**
>
> Thank you for your thorough and constructive feedback. We have made substantial improvements to address your concerns about clarity, methodology, and presentation:
>
> - We significantly expanded Section 3 to provide a clearer and more detailed explanation of our analyses
> - To ensure better transparency, we added Appendix A, which provides comprehensive documentation of our experimental methodology and data processing
> - We also added Appendix B, which presents baseline statistical analyses of the human evaluation data that we collected
> - Based on your valuable feedback about visualization clarity, we developed three new figures (3, 5, and 7) that more effectively communicate our results
> - We also removed potentially confusing visualizations and enhanced (but forgot to update) Figure 6's legibility to better support our key findings
>
> While we understand your suggestion about additional models, we have neither time nor money to change this, and we believe our enhanced methodology and clearer presentation merit your reassessment of our paper.

---

### Official Review · Reviewer_sJ5t · 2024-11-04

**Soundness:** 4
**Presentation:** 4
**Contribution:** 4
**Rating:** 8
**Confidence:** 5

**Summary:**

The paper initially explores the correlation between NLP benchmarks and human evaluation. With the advent of increasingly capable LLMs, human evaluations have become a steady and major alternative choice to evaluate the efficacy, performance and capabilities of LLMs. An important question that generally arises with the choice is whether NLP benchmarks are useless since human evaluations are costly and time consuming and are not always a gold standard. Where do NLP benchmarks fall? This paper explores this question and also explores the possibility of predicting human evaluations from NLP benchmarks.

Two key questions are asked:
- To what extent are human evaluations and NLP benchmarks correlated?
- How well can benchmarks predict expensive and time-intensive human evaluations?

The researchers use all the Llama chat-2 models (7,13,30 and 70B parameters) to establish this, which were trained on 2T tokens and fine-tuned using SFT and RLHF. Human evaluations are collected by evaluating the Llama2 chat models pairwise against ChatGPT 3.5 on dataset a of single-turn and multi-turn prompts, where responses are sampled from each model. 3 Human annotators independently provide a pairwise comparison on the Likert scale (1 to 7, where 1 means chat llama preferred and 7 means chatgpt 3.5 preferred). uThey end up doing a large-scale study spanning factual questions, language assistance, writing, procedural questions, reasoning and many more.  The Chat Llama 2 models are evaluated on many popular NLP benchmarks right from AGI Eval, Ai2 Reasoning Challenge, Big Bench Hard, Boolq, commonseqa, GSM8k, MMLU, MATH, QuAC, PiQA and many more. Standard evaluation processes are used.

The findings revealed that NLP benchmarks are broadly highly correlated with human evaluations, with certain benchmarks showing particularly strong correlations. The most predictive benchmarks included specific subsets of MMLU (covering topics like nutrition, human aging, and sociology), portions of BIG Bench Hard, HellaSwag, ARC, RACE, PIQA, Natural Questions, QuAC, and CommonSenseQA. However, some benchmarks showed weaker correlations, including ETHOS, Kth Sentence, most of Inverse Scaling, OpenBookQA, COPA, SciBench, and SIQA.

Using overparameterized linear regression, the researchers successfully demonstrated that NLP benchmark scores could predict human evaluation scores with reasonable accuracy. Despite the small sample size of only four models, leave-one-out cross-validation showed promising results, suggesting that faster and cheaper NLP benchmarks might effectively predict slower, more expensive human evaluations in many cases.

The authors note several limitations, including the small sample size, the assumption of linearity in their predictive models, and potential limits to generalizability across different model families thus rounding up the study and paving the way for future work as well.

**Strengths:**

1] The question at the center of the paper -- "Correlation between NLP benchmarks and Human Evaluations," is an important central question to NLP evaluation in general. Human Evaluations are considered (somewhat so) the gold standard of evaluation but are extremely time-consuming and expensive to run; as models get more capable the human evaluations also get even more costlier because now we require experts to evaluate vs requiring less advanced folks earlier, but we can reliably construct more difficult benchmarks for models, so if these two things are correlated, perhaps lesser focus can be placed on human evaluations.

2] Predicting Human Evaluations is a difficult task, and LLMs as judges are being increasingly explored as an alternative to human evaluations. The method in the paper also showcases some important insights into this process.

**Weaknesses:**

1] The small sample size brings into question the generalizability of these insights and results.

2] Only uses GPT-3.5 as the comparative model, no insight is provided into why this is the case? And also lacks any discussion of whether chatgpt 3.5 is a reasonable choice of a baseline.

3] Perhaps a granular analysis of what makes a benchmark more correlated? Is there something common in the correlated benchmarks? This would also pave the way to designing and determining better benchmarks.

**Questions:**

1) Why chatgpt 3.5? Could you justify the choice of this model? Why was chatgpt 3.5 the model chosen for comparison, is it a reasonable choice for a baseline?

2) Could you generally talk about the distribution of the Likert scale that you got from the pairwise evals? Was there anything at all in which chatgpt was substantially better and generally chosen? (Assumption here that I suppose llama-2 would be usually better than chatgpt 3.5 in all cases)

3)  if these outputs were obtained from Chatgpt 3.5, which API was it received from, and what was the exact cutoff (e.g., ChatGPT-3.5-0604, etc.)?

4) Pairwise evals ultimately show revealed preferences and model choice between two outputs. Do you think this translates to human evaluation directly on model outputs (not comparisons) on NLP parameters like coherence, semantic relevance, factual relevance, etc.? Could you comment on the choice of pairwise evals?

5) Just a general question about related work: is there no related work? (while this correlation aspect might not have been explicitly studied), Studies have considered which of them is better in MT, summarization, and other NLP areas. Can you provide a more comprehensive overview of related work, including studies that have compared human evaluations and benchmarks in specific NLP tasks like MT/Summarization, and contextualize this work in the broader field?

6) Could you conduct a detailed analysis of features/characteristics shared by highly correlated benchmarks? I think that would help a lot in designing benchmarks in the future.

---

> ### Author Response · Authors · 2024-11-22
> **Response to Reviewer sJ5t**
>
> Thank you for reviewing our paper! We are grateful to read that you find the question at the heart of our paper is an important and central question to NLP evaluations, and that our work contributes important insights into these evaluations.
>
> To address a subset of the concerns you raised:
>
> > The small sample size brings into question the generalizability of these insights and results.
>
> We agree. Sadly, the human experiments are costly and slow and we ran into errors collecting human evaluations for additional models. We highlight this as a core limitation of our work but see no way around it.
>
> > Only uses GPT-3.5 as the comparative model, no insight is provided into why this is the case? And also lacks any discussion of whether chatgpt 3.5 is a reasonable choice of a baseline.
>
> > Why chatgpt 3.5? Could you justify the choice of this model? Why was chatgpt 3.5 the model chosen for comparison, is it a reasonable choice for a baseline?
>
> We used GPT-3.5 because at the time this data was collected, GPT-3.5 was a good balance of three desirable properties for our study: (i) performant, (ii) cheap, and (iii) stable.
>
> We feel the choice of baseline is not so critical because our goal is to assess how improvements in NLP benchmark scores correlate with and predict improvements in human evaluation scores. Thus, what matters is how models vary/improve. We acknowledge that multiple baselines would be ideal, but this was out of budget.
>
> > if these outputs were obtained from Chatgpt 3.5, which API was it received from, and what was the exact cutoff (e.g., ChatGPT-3.5-0604, etc.)?
>
> Line 110: gpt-3.5-turbo-0301.
>
> > Perhaps a granular analysis of what makes a benchmark more correlated? Is there something common in the correlated benchmarks? This would also pave the way to designing and determining better benchmarks.
>
> This could be quite an interesting analysis! However, this would be beyond the scope of our paper. This would require trying to “featurize” benchmarks and then testing which features of benchmarks lead to correlations between benchmarks.
>
> > Just a general question about related work: is there no related work? (while this correlation aspect might not have been explicitly studied), Studies have considered which of them is better in MT, summarization, and other NLP areas. Can you provide a more comprehensive overview of related work, including studies that have compared human evaluations and benchmarks in specific NLP tasks like MT/Summarization, and contextualize this work in the broader field?
>
> This is indeed a shortcoming that we will address. We will add a Related Work section.
>
> We will update the manuscript in 1-2 days and address your remaining concerns then.

---

> > ### Author Response · Authors · 2024-11-26
> > **Follow Up to Reviewer sJ5t**
> >
> > Dear Reviewer sJ5t,
> >
> > To circle back to your review,
> >
> > > Just a general question about related work: is there no related work?
> >
> > As promised, we added a Related Work section in the revised manuscript. We are continuing to work on other improvements and will have an updated manuscript in a day.
> >
> > > The small sample size brings into question the generalizability of these insights and results.
> >
> > To reiterate, we do agree. Our sample size is small because collecting human evaluations for a single model costs ~$250k USD and we ran into errors trying to collect human evaluations of other models. We are happy to highlight this limitation, but we sadly have no way of fixing it.

---

> > > ### Comment · Reviewer_sJ5t · 2024-11-26
> > > **Follow up**
> > >
> > > I thank the authors for their thoughtful and prompt responses. I believe I have already scored at a level that marks the importance of the work, but any higher wouldn't be merited given its scope.
> > >
> > > The related work is done well, and I would suggest a few nits that would be of great help if the choice of gpt3.5, as you have answered, is also written in the paper and motivates the approach given the backdrop of high costs. Maybe there are other ways to explore the generalizability of this work, even with limited samples, and I hope you explore them.  Thank you for writing the paper.

---

### Meta-Review · Area_Chair_E6kR · 2024-12-18

**Metareview:**

This paper studies the relationships between human evaluations and NLP benchmarks. They find that most NLP benchmarks are broadly highly correlated with human evaluations, and they also fit models to predict a language model’s human evaluation scores from academic evaluation scores. The problem studied in this paper is interesting and important. However, the paper is not well written, and the reviewers raised many writing issues of the paper. The study only used four LLMs, which is not enough. In summary, this paper needs to be greatly improved and more LLMs need to be added to the study.

**Additional Comments On Reviewer Discussion:**

Reviewers who gave negative scores were not satisfied with the rebuttal and did not change their scores.

---

### Decision · Program_Chairs · 2025-01-22

Reject